# The individuality of shape asymmetries of the human cerebral cortex

Yu-Chi Chen[1,2,3]*, Aurina Arnatkevičiūtė[1], Eugene McTavish[1,2,4], James C Pang[1,2], Sidhant Chopra[1,2,5], Chao Suo[1,2,6], Alex Fornito[1,2†], Kevin M Aquino[1,2,7,8,9†], for the Alzheimer's Disease Neuroimaging Initiative

[1]Turner Institute for Brain and Mental Health, School of Psychological Sciences, Monash University, Melbourne, Australia; [2]Monash Biomedical Imaging, Monash University, Melbourne, Australia; [3]Monash Data Futures Institute, Monash University, Melbourne, Australia; [4]Healthy Brain and Mind Research Centre, Faculty of Health Sciences, Australian Catholic University, Fitzroy, Australia; [5]Department of Psychology, Yale University, New Haven, United States; [6]BrainPark, Turner Institute for Brain and Mental Health, School of Psychological Sciences, Monash University, Melbourne, Australia; [7]School of Physics, University of Sydney, Sydney, Australia; [8]Center of Excellence for Integrative Brain Function, University of Sydney, Sydney, Australia; [9]BrainKey Inc, San Francisco, United States

**Abstract** Asymmetries of the cerebral cortex are found across diverse phyla and are particularly pronounced in humans, with important implications for brain function and disease. However, many prior studies have confounded asymmetries due to size with those due to shape. Here, we introduce a novel approach to characterize asymmetries of the whole cortical shape, independent of size, across different spatial frequencies using magnetic resonance imaging data in three independent datasets. We find that cortical shape asymmetry is highly individualized and robust, akin to a cortical fingerprint, and identifies individuals more accurately than size-based descriptors, such as cortical thickness and surface area, or measures of inter-regional functional coupling of brain activity. Individual identifiability is optimal at coarse spatial scales (~37 mm wavelength), and shape asymmetries show scale-specific associations with sex and cognition, but not handedness. While unihemispheric cortical shape shows significant heritability at coarse scales (~65 mm wavelength), shape asymmetries are determined primarily by subject-specific environmental effects. Thus, coarse-scale shape asymmetries are highly personalized, sexually dimorphic, linked to individual differences in cognition, and are primarily driven by stochastic environmental influences.

*For correspondence: yu-chi.chen@monash.edu

†These authors contributed equally to this work

## Editor's evaluation

The article is of interest to scientists who study neuroanatomy or the many behavioral phenotypes that have been proposed be associated with left–right asymmetry of the human brain. The methodology is sophisticated and rigorously applied.

## Introduction

Asymmetries in brain structure and function are found throughout the animal kingdom (*Duboc et al., 2015*; *Güntürkün et al., 2020*; *Corballis and Häberling, 2017*; *Güntürkün and Ocklenburg, 2017*) and can be discerned at multiple spatial scales, ranging from differences in the size and shape of the cerebral hemispheres through measures of regional morphometry and connectivity to cellular and molecular organization (*Güntürkün et al., 2020*; *Güntürkün and Ocklenburg, 2017*; *Esteves et al.,*

2021). At the coarsest scale, the most salient feature of anatomical asymmetry in the human brain is cerebral torque, in which the right hemisphere appears to be warped in the rostral direction relative to the left hemisphere (*Li et al., 2018*; *Toga and Thompson, 2003*; *Zhao et al., 2021*). More fine-grained asymmetries of specific sulci/gyri (*Kang et al., 2015*) and brain regions *Kong et al., 2018*; *Plessen et al., 2014* have also been investigated. For example, the superior temporal sulcus, which is adjacent to the Wernicke's area, shows a leftward asymmetry in length (*Gómez-Robles et al., 2013*).

Asymmetries in brain organization are often considered at an average level across a population of individuals (*Toga and Thompson, 2003*; *Kong et al., 2018*; *Kong et al., 2022*; *Deep-Soboslay et al., 2010*; *Postema et al., 2019*). These population-based asymmetry features have been studied extensively and are thought to have important implications for both functional lateralization and abnormal brain function in a wide range of psychiatric and neurological diseases (*Esteves et al., 2021*; *Plessen et al., 2014*; *Postema et al., 2019*; *Cai et al., 2015*; *Fling et al., 2014*). For example, the planum temporale of the left hemisphere, which encompasses Wernicke's area, has been consistently shown to be larger than the right for most healthy individuals (*Güntürkün et al., 2020*; *Toga and Thompson, 2003*; *Royer et al., 2015*; *Takao et al., 2011*), and patients with schizophrenia often show reduced leftward asymmetry in planum temporale compared to healthy individuals (*Clark et al., 2010*; *Ratnanather et al., 2013*; *Corballis, 2013*). However, many findings with respect to asymmetries of specific brain regions have been inconsistent in terms of the directions and magnitudes of asymmetry observed (*Kong et al., 2018*; *Plessen et al., 2014*; *Kong et al., 2022*). The correlates of these asymmetries are also unclear (*Güntürkün and Ocklenburg, 2017*; *Plessen et al., 2014*; *Kong et al., 2022*; *Kurth et al., 2018*; *Núñez et al., 2018*). For example, two fundamental characteristics often examined in relation to cerebral asymmetry are sex and handedness. Some studies have found that the surface area (*Kong et al., 2018*), shape (*Núñez et al., 2018*; *Wachinger et al., 2015*; *Kovalev et al., 2003*), volume (*Guadalupe et al., 2015*), and torque (*Zhao et al., 2021*) of cortical structures in males are more asymmetric than in females, whereas other studies have found no sex differences (*Takao et al., 2011*; *Narr et al., 2007*). Similarly, some studies have found associations between cerebral asymmetry and handedness (*Zhao et al., 2021*; *Deep-Soboslay et al., 2010*; *Steinmetz et al., 1991*), with others reporting no such effect (*Kong et al., 2018*; *Plessen et al., 2014*; *Wachinger et al., 2015*; *Narr et al., 2007*; *Good et al., 2001*; *Guadalupe et al., 2014*; *Maingault et al., 2016*).

Some of these inconsistencies may arise from the disparate methodologies and the heterogeneous nature of the brain asymmetries across the population (*Toga and Thompson, 2003*; *Kong et al., 2018*; *Gómez-Robles et al., 2013*; *Kong et al., 2022*; *Deep-Soboslay et al., 2010*; *Postema et al., 2019*). Despite some consistent asymmetry features across the population (*Güntürkün et al., 2020*; *Toga and Thompson, 2003*; *Royer et al., 2015*; *Takao et al., 2011*), there is also considerable individual variability around population means, with many people often showing little or even reversed asymmetries relative to the prevalent pattern of the population (sometimes also referred to as antisymmetry) (*Corballis and Häberling, 2017*; *Gómez-Robles et al., 2013*; *Gómez-Robles et al., 2016*; *Neubauer et al., 2020*). The distinction between population-level and individual-specific asymmetries is essential as they are thought to arise from distinct mechanisms (*Gómez-Robles et al., 2016*; *Sherwood and Gómez-Robles, 2017*). Populational-level asymmetries are hypothesized to have a genetic basis (*Zhao et al., 2021*; *Kong et al., 2018*; *Gómez-Robles et al., 2016*; *Neubauer et al., 2020*; *Sherwood and Gómez-Robles, 2017*; *Francks, 2015*; *de Kovel et al., 2018*; *Graham and Özener, 2016*; *Sha et al., 2021*), whereas individual-specific asymmetries, which describe the way in which a given individual departs from the population mean, may reflect environmental influences, developmental plasticity, or individual-specific genetic perturbations (*Gómez-Robles et al., 2016*; *Neubauer et al., 2020*; *Sherwood and Gómez-Robles, 2017*; *Francks, 2015*; *de Kovel et al., 2018*; *Graham and Özener, 2016*; *Nadig et al., 2021*). Notably, cortical asymmetries of the human brain are more variable across individuals than other primates at both regional and global hemispheric levels (*Gómez-Robles et al., 2013*; *Neubauer et al., 2020*). The variability is most evident in regions of heteromodal association cortex, leading some to conclude that high levels of variability in asymmetry may have emerged in line with the evolution of human-specific cognition (*Gómez-Robles et al., 2013*), although the relationship between the asymmetries of the human brain and individual differences in cognition is still largely unknown.

Traditional analysis methods, which rely on standard image processing techniques such as image registration and spatial smoothing, minimize individual variation and thus have limited sensitivity for

studying individual-specific asymmetries (*Gomez-Robles et al., 2018*; *Wachinger et al., 2016*). Moreover, most past studies have focused on morphological properties related to the size of specific brain regions, such as estimates of gray matter volume, cortical thickness, or surface area, often measured at fine-grained resolutions, such as individual voxels or the vertices of cortical surface mesh models (*Cai et al., 2015*; *Takao et al., 2011*; *Kurth et al., 2018*; *Good et al., 2001*; *Maingault et al., 2016*; *Kruggel and Solodkin, 2020*; *Kurth et al., 2015*). Many of the most obvious features of cerebral asymmetry arise from variations in brain shape, which are not captured by size-related descriptors (*Wachinger et al., 2015*; *Reuter et al., 2009*). Indeed, it is possible for two objects to have identical volume but have very different shapes (*Reuter et al., 2009*; *Ge et al., 2016*). In addition, shape variations can occur at different spatial resolution scales, from the presence and configuration of specific sulci at fine scales to more global patterns such as cerebral petalia at coarser scales. Conventional analyses only consider the finest resolvable scale (i.e., point-wise differences) and have limited sensitivity for identifying important morphological variations that occur over large swathes of cortical tissue.

A comprehensive, multiscale description of cortical shape, from the finest to coarsest scales, can be derived through a spectral analysis of cortical geometry based on solutions to the Helmholtz equation (*Wachinger et al., 2015*; *Reuter et al., 2009*; *Reuter et al., 2006*), which is fundamental in many branches of physics, engineering, chemistry, and biology (*Lévy, 2006*). The equation can be solved by formulating it as an eigenfunction–eigenvalue problem of the Laplace–Beltrami operator (LBO) (see 'Materials and methods'). Importantly, the characteristics of the eigenfunctions and eigenvalues depend on the cortical shape for which the equation is solved (*Reuter et al., 2006*; *Lévy, 2006*), and thus, the spectral analysis provides a comprehensive description of the intrinsic geometry of a given object, akin to a 'Shape-DNA' (see 'Materials and methods'; *Reuter et al., 2006*). The application of such Shape-DNA analysis to human magnetic resonance imaging (MRI) data has shown that shape properties of cortical and subcortical structures have superior sensitivity compared to traditional, size-based measures for identifying individual subjects (*Wachinger et al., 2015*), classifying and predicting the progress of psychiatric and neurological diseases (*Wachinger et al., 2016*; *Richards et al., 2020*), and detecting genetic influences on brain structure (*Ge et al., 2016*; *Wachinger et al., 2018*). However, a detailed characterization of individual-specific asymmetries in cerebral shape is lacking.

Here, we introduce methods for constructing an individual-specific measure of cortical asymmetry, called the shape asymmetry signature (SAS; see 'Materials and methods'). The SAS characterizes pure shape asymmetries of the whole cortical surface, independent of variations in size, across a spectrum of spatial scales. We apply this methodology to three independent longitudinal datasets to test the hypothesis that cortical shape asymmetry is a highly personalized and robust feature that can identify individuals, akin to a cortical asymmetry fingerprint. We then use the identifiability values to identify optimal spatial scales at which robust individual differences are most salient. We also compare the identifiability of the SAS and shape descriptors of individual hemispheres, asymmetries in traditional size-based descriptors, or patterns of inter-regional functional connectivity (so-called connectome fingerprinting; *Finn et al., 2015*) to test the hypothesis that the SAS is a more individually unique property of brain organization than unihemispheric and functional properties. We further elucidate the relationships between the SAS and sex, handedness, as well as cognitive performance across multiple tasks. Finally, we test the hypothesis that individual-specific asymmetry features are largely driven by environmental influences using classical heritability modeling of twin data.

## Results
### Cortical shape asymmetries are individually unique

To understand how cortical shape asymmetries vary across individuals, we examined the degree to which different cortical shape descriptors (defined below) can be used to identify individual brains from a large sample of T1-weighted MRIs. We analyzed healthy subjects from three open-source datasets – the latest Open Access Series of Imaging Studies (OASIS-3; *LaMontagne et al., 2019*), the Human Connectome Project (HCP; *Van Essen et al., 2013*), and the Alzheimer's Disease Neuroimaging Initiative (ADNI; https://ida.loni.usc.edu/) – in which individuals had at least two anatomical MRI scans acquired at different time points (separated by 1 day to several years; see 'Materials and methods'). For each dataset, we asked whether the shape descriptors for an individual estimated

from the first scan could accurately identify the same participant's second scan. Within each dataset, the shape descriptor was calculated from the cortical surfaces at the white and gray matter boundary estimated either from FreeSurfer (*Fischl et al., 2002*) (OASIS-3 and ADNI) or FreeSurfer-HCP (HCP), which is a FreeSurfer (*Fischl et al., 2002*) pipeline with some HCP-specific enhancements (*Glasser et al., 2013*). Shape-DNA (*Reuter et al., 2009*; *Reuter et al., 2006*) analysis was employed to obtain multidimensional shape descriptors for each hemisphere that quantify the shape of each individual's cortex, as defined by the eigenvalue spectrum of the LBO (*Figure 1A and B*; see 'Materials and methods'). Each eigenvalue is associated with a corresponding eigenfunction, which describes shape variations at a particular spatial wavelength, ordered from coarse to fine-grained scales (*Figure 1B*). These eigenfunctions are orthogonal by construction and thus represent a basis set for cortical shape variations much like the sinusoidal basis used in Fourier decomposition of signals, with the corresponding eigenvalue being analogous to the wave frequency at each spatial scale. Critically, we normalized the surface area (*Reuter et al., 2009*) of the meshes prior to Shape-DNA analysis to ensure that the resulting eigenvalue spectra were independent of individual differences in brain size (see 'Materials and methods').

To investigate the uniqueness of these shape descriptors to individual brains, we performed an identifiability analysis (*Amico and Goñi, 2018*; *Mansour L et al., 2021*), where identifiability was quantified as the degree to which the surface eigenvalue spectrum of an individual at scan time 1 was more similar to the same person's spectrum at time 2, relative to other people's time 2 spectra (*Figure 1C*; see also 'Materials and methods'). To determine whether identifiability is maximized at any specific scales, we repeated the analysis multiple times, initially by taking only the first two eigenvalues, which describe shape variations at the coarsest scale, and then incrementally adding eigenvalues representing more fine-grained features to a maximum of 1000. Plotting the identifiability score as a function of the number of eigenvalues allows us to identify characteristic spatial scales at which the identifiability score is maximized (*Figure 1D*). In other words, it allows us to identify the scales at which individual-specific shape features are most pronounced. We repeated this procedure using the eigenvalue spectra for the left and right hemispheres alone, the combination of both (which describes the shape of both hemispheres), and for the SAS, which quantifies shape asymmetries as the difference between the left and right hemisphere eigenvalue spectra (see *Figure 2* for details). Finally, we utilized the spatial scales with maximum identifiability (*Figure 1D*) to examine the relationships between the SAS and sex, handedness, cognition, and heritability. In general, a brain with a higher degree of shape asymmetry has SAS values that more strongly depart from zero (*Figure 1—figure supplement 1*).

*Figure 2A–C* shows the identifiability scores obtained for the different shape descriptors. In all three datasets, across a broad range of spatial scales, identifiability was highest for the SAS, followed by the combination of left and right hemisphere eigenvalues, and then each hemisphere alone. This result indicates that individual variability in the asymmetry of cortical shape is greater than the variability of shape across the whole cortex or within each hemisphere alone. *Figure 2A–C* also shows identifiability scores obtained when trying to identify an individual's left hemisphere using right hemisphere shape descriptors obtained at the same time point. These scores are very low, indicating that shape variations between the two hemispheres are largely independent of each other and lack a consistent pattern amongst subjects. In other words, for any given person, the shape of one hemisphere offers little individually unique information about the shape of the other hemisphere.

## Individually unique variations of cortical shape asymmetry are maximal at coarse spatial scales

We next investigated the scale-specificity of SAS identifiability. *Figure 2A–C* shows that SAS identifiability sharply increases to a peak as we use more eigenvalues to characterize the surface shape at finer scales (i.e., as we add more shape information from finer spatial scales), before gradually falling again. This peak identifies a characteristic spatial scale in which individual differences in shape asymmetries are maximally unique (see also *Figure 2—figure supplement 1*).

Peak SAS identifiability was observed using the first 126 and 122 eigenvalues for the OASIS-3 (*Figure 2A*) and ADNI (*Figure 2B*) data, respectively. At these scales, the subject identifiability scores were 4.93 (p=0; estimated by permutation; see 'Statistical analysis' section for details) for OASIS-3 and 5.03 (p=0) for ADNI. For the HCP data, peak SAS identifiability was observed when using the first

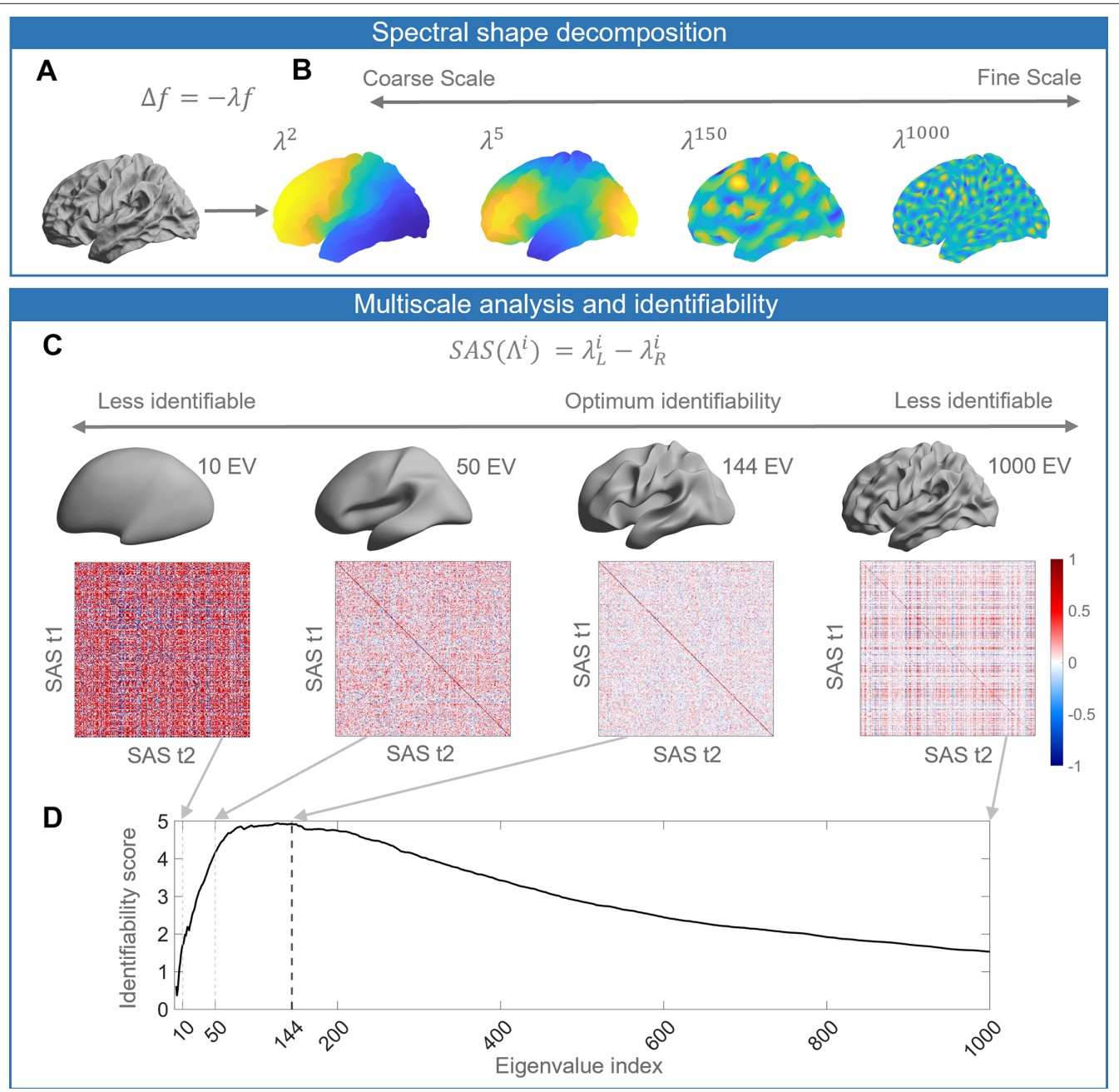

**Figure 1.** Schematic of our analysis workflow. (**A**) The shapes of the left and right hemispheres are independently analyzed using the Laplace–Beltrami operator (LBO) via the Shape-DNA algorithm (***Reuter et al., 2009***; ***Reuter et al., 2006***). (**B**) Eigenfunctions of the LBO are obtained by solving the Helmholtz equation on the surface, given by $\Delta f = -\lambda f$, where $f$ corresponds to a distinct eigenfunction, and $\lambda$ is the corresponding eigenvalue. Each eigenvalue $\lambda^i, i = 1, 2, \ldots, 1000$, quantifies the degree to which a given eigenfunction is expressed in the intrinsic geometry of the cortex. Higher-order eigenvalues describe shape variations at finer spatial scales. (**C**) The shape asymmetry signature (SAS) is quantified as the difference in the left and right hemisphere eigenvalue spectra, providing a summary measure of multiscale cortical shape asymmetries. To investigate the identifiability of the SAS, we use Pearson's correlation to calculate the similarity between the SAS vectors obtained for the time 1 (t1) and time 2 (t2) two scans from the same individuals (diagonal elements of the matrices) as well as the correlation between t1 and t2 scans between different subjects (off-diagonal elements). We estimate identifiability by first correlating the initial two eigenvalues, then the initial three eigenvalues, and so on to a maximum of 1000 eigenvalues. Here, we show examples of correlation matrices obtained when using the first 10, 50, 144, and 1000 eigenvalues, and the cortical surface reconstructions show the shape variations captured by corresponding spatial scales. (**D**) Repeating the identifiability analysis up to a maximum of 1000 eigenvalues yields a curve with a clear peak, representing the scale at which individual differences in cortical shape are maximal. For the SAS, this peak occurs when the first 144 eigenvalues are used (black dashed line), which offers a fairly coarse description of shape variations (see panel **C**). We then use a similar

*Figure 1 continued on next page*

*Figure 1 continued*

analysis approach to investigate associations between scale-specific shape variations and sex, handedness, cognitive functions, as well as heritability. The data in this figure are from the OASIS-3 (n = 233) cohort, and the cortical surfaces are from a population-based template (fsaverage in FreeSurfer).

The online version of this article includes the following figure supplement(s) for figure 1:

**Figure supplement 1.** Higher shape asymmetry signature (SAS) values characterize brains with stronger cortical shape asymmetries.

268 eigenvalues (identifiability score = 6.74; p=0; *Figure 2C*), but the identification curve flattened after the first 137 eigenvalues (identifiability score = 6.56), which is closely aligned with the OASIS-3 and ADNI datasets.

In the case of a perfect sphere, the shape spectral analysis yields subsets of degenerate eigenvalues with equal magnitude (*Robinson et al., 2016*), within which the corresponding eigenfunctions represent orthogonal rotations of the same spatial pattern at a given scale. For example, eigenfunctions 2–4 of a sphere represent coarse-scale gradients in the anterior–posterior, inferior–superior, and left–right axes. As the cortex is topologically equivalent to a sphere, the spherical eigen-groups offer a natural way to identify characteristic spatial scales, more succinctly summarize cortical shape variations (*Robinson et al., 2016*), and smooth out eigenvalue-specific fluctuations at a given scale (see 'Materials and methods'). We averaged the identifiability scores for each harmonic group and plotted these as a function of the group index in *Figure 2D–F*. The group mean identifiability score peaks at the 11th eigenvalue group for the OASIS-3 (mean identifiability score = 4.93) and ADNI (mean identifiability score = 5.06) datasets, which is comprised of the first 144 eigenvalues. Identifiability also reaches a near-plateau at the 11th group for the HCP data (mean identifiability score = 6.47), with an additional marginal increase observed at the 16th group (mean identifiability score = 6.69). Thus, the first 144 eigenvalues represent a stable and robust characteristic scale at which individual uniqueness in cortical shape asymmetry is strongest. The 11th group corresponds to a wavelength of approximately 37 mm in the case of the population-based template (fsaverage in FreeSurfer; *Supplementary file 1* shows the corresponding wavelengths of the first 14 eigen-groups; *Figure 2G* shows the spatial scales corresponding to the cumulative eigen-groups).

A reconstruction of the cortical surface using the first 144 eigenfunctions is shown in *Figure 2H*. The reconstruction captures shape variations at a coarse scale, representing major primary and secondary sulci, but with minimal additional details. If we include additional eigenfunctions to capture more fine-scale anatomical variations, inter-session image differences increase, suggesting that finer spatial scales may be capturing dynamic aspects of brain structure that are more susceptible to increased measurement noise (*Figure 2—figure supplement 2*). This same characteristic scale was obtained after repeating the identifiability analysis over the longest inter-scan intervals in the ADNI and OASIS-3 datasets (*Figure 2—figure supplement 3*), indicating that our results are robust over time windows ranging from 1 day to more than 6 years.

## Shape asymmetries are more identifiable than classical morphological and functional measures

We next compared the identifiability of the SAS to scores obtained using asymmetries in classical morphological descriptors such as regional surface area, cortical thickness, and gray matter volume, and measures of inter-regional functional connectivity (*Figure 3*), which have previously been shown to yield high identifiability (*Finn et al., 2015*; *Amico and Goñi, 2018*). Identifiability scores obtained with the SAS were much higher than those obtained by regional asymmetries in size-related morphological measures with the HCP-MMP1 atlas (*Glasser et al., 2016*; *Figure 3A and B*). We also found that SAS identifiability was higher when using our surface area normalization procedure compared to the SAS computed without this procedure (*Figure 3—figure supplement 1*; see 'Materials and methods'). Since the normalization isolates the pure effects of shape independent of brain size, the results converge to indicate that individual variability in brain anatomy is higher when considering asymmetries in cortical shape compared to more traditional size-based morphological descriptors.

*Figure 3C–F* compares the identifiability scores obtained from the SAS to those obtained using inter-regional functional connectivity (see 'Materials and methods'), within the HCP test–retest data. Functional connectivity was quantified for the entire cortex using four different regional parcellations defined at different spatial scales (Schafer 100, Schaefer 300, Schaefer 900 [*Schaefer et al.,*

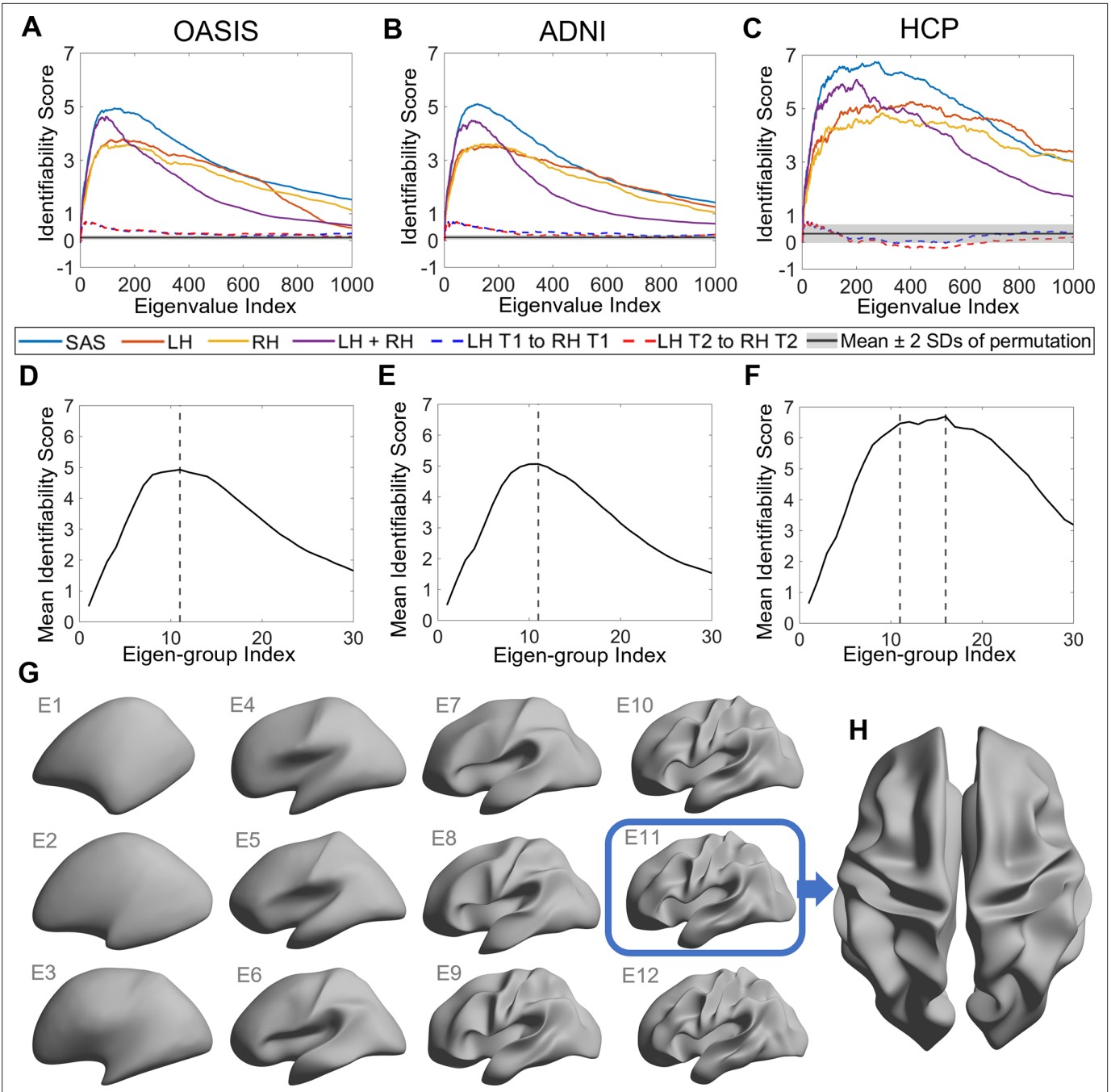

**Figure 2.** Identifiability of different shape descriptors at different spatial scales. (**A–C**) Identifiability scores for shape features across eigenvalue indices. The identifiability scores of the shape asymmetry signature (SAS) are generally higher than the scores for shape descriptors of individual hemispheres or scores obtained when concatenating both hemispheres across three datasets (OASIS-3: n = 233; ADNI: n = 208; HCP test–retest: n = 45). The SAS scores are also much higher than the scores obtained by randomly shuffling the order of the subjects at time 2 (shaded area represents mean ± 2 SDs). (**D–F**) The cumulative mean identifiability scores for each eigenvalue group, derived from correspondence with spherical harmonics (**Robinson et al., 2016**). The peak mean identifiability occurs at the 11th eigenvalue group for the OASIS-3 (**D**) and ADNI data (**E**), representing the first 144 eigenvalues. The curve of the mean identifiability score for the HCP test–retest data (**F**) flattens after the 11th group and peaks at the 16th group. (**G**) Cortical surfaces reconstructed at different spatial scales, starting with only the first eigen-group (E1) and incrementally adding more groups to a maximum of the first 12 eigen-groups (E12). (**H**) Overhead view of the spatial scale corresponding to the eigen-group at which identifiability is maximal in the OASIS-3 and ADNI datasets (i.e., the first 11 eigen-groups, corresponding to the first 144 eigenvalues).

The online version of this article includes the following figure supplement(s) for figure 2:

**Figure supplement 1.** Understanding the identifiability score.

*Figure 2 continued on next page*

*Figure 2 continued*

**Figure supplement 2.** Inter-session variability in cortical shape is higher at more fine-grained spatial scales.

**Figure supplement 3.** Subject identifiability scores recalculated for data from MRI sessions with the longest inter-sessional interval.

*2018*], and HCP-MMP1 [*Glasser et al., 2016*] atlas). The SAS outperformed all functional identifiability scores, indicating that cortical shape shows greater specificity to an individual than patterns of functional connectivity.

## Cortical shape asymmetries are related to sex but not handedness

Sex and handedness are two characteristics that have frequently been examined in relation to brain asymmetry (*Güntürkün et al., 2020*; *Toga and Thompson, 2003*; *Kong et al., 2018*; *Plessen et al., 2014*; *Wachinger et al., 2015*; *Guadalupe et al., 2015*; *Narr et al., 2007*; *Good et al., 2001*; *Kong et al., 2021*). We used a general linear model (GLM) with permutation testing and accounting for familial structure (*Winkler et al., 2016*; *Winkler et al., 2015*) of the HCP data to evaluate the association between these two characteristics and the SAS defined at each eigenvalue ranging between the 2nd and 144th. After false discovery rate (FDR) correction, males and females showed significant differences in asymmetry scores for the 2nd ($P_{FDR}$ = 0.037), 6th ($P_{FDR}$ = 0.037), 8th ($P_{FDR}$ = 0.039), 52nd ($P_{FDR}$ = 0.030), and 84th ($P_{FDR}$ = 0.037) eigenvalues (*Figure 4A*), where female brains showed more rightward asymmetry than males in these eigenvalues. These five eigenvalues come from four different eigen-groups, and the corresponding spatial scales of these eigenvalues are shown in *Figure 4B*. These eigenvalues relate to shape variations over coarse scales. For instance, for the 2nd eigenvalue ($L$ = 1; see 'Materials and methods' for the definition of $L$), the wavelength is of order 300 mm, which is about half the circumference of the brain; for the most-fine grained eigenvalue, the 84th eigenvalue ($L$ = 9), the wavelength is about 44 mm. We note however that the sex differences are small, with considerable overlap between male and female distributions (*Figure 4A*). No such effects of handedness on the SAS surpassed the FDR-corrected threshold. We also found that the overall asymmetry level (i.e., the sum of the SAS) was not correlated with either handedness or sex.

## Individual differences in cortical shape asymmetry correlate with cognitive functions

We used canonical correlation analysis (CCA) (*Winkler et al., 2020*) to examine associations between the SAS and 13 cognitive measures from the HCP dataset (n = 1094; see 'Materials and methods') selected to represent a wide range of cognitive functions (*Kong et al., 2019*; see 'Materials and methods' for details). To reduce the dimensionality of the SAS measures and ensure equivalent representation of asymmetries at each spatial scale, we took the mean SAS value for each of the 1st to 11th eigen-groups, spanning the 2nd to 144th eigenvalues. To minimize collinearity of the cognitive variables, we applied principal component analysis (PCA) to the 13 cognitive measures and retained the first four principal components (PCs), which explained 80% of the variance. The analysis revealed a single statistically significant canonical mode (CCA r = 0.187; $P_{FWER}$ = 0.032; *Figure 5A*). *Figure 5B* shows that the mode has significant positive loadings from mean SAS scores in eigen-groups 2, 4, 5, and 11, and significant negative loadings from eigen-groups 3, 6, 7, and 10. *Figure 5C* indicates that 12 of the 13 cognitive measures showed significant positive correlations with the canonical variate, indicating that it captures covariance with general cognitive ability. Thus, our findings identify strong scale-specificity of associations between cortical shape asymmetry and cognition, with a greater leftward asymmetry in scales captured by eigen-groups 2 (~170 mm wavelength), 4 (~95 mm wavelength), 5 (~75 mm wavelength), and 11 (~37 mm wavelength) being associated with better performance across most cognitive measures, and a greater leftward asymmetry in scales captured by eigen-groups 3 (~120 mm wavelength), 6 (~65 mm wavelength), 7 (~55 mm wavelength), and 10 (~40 mm wavelength) being associated with poorer cognitive performance.

## Cortical shape asymmetries are primarily driven by unique environmental influences

To characterize genetic and environmental effects on cortical shape and its asymmetry, we calculated the heritability of each eigenvalue within the left and right hemispheres, as well as for the SAS.

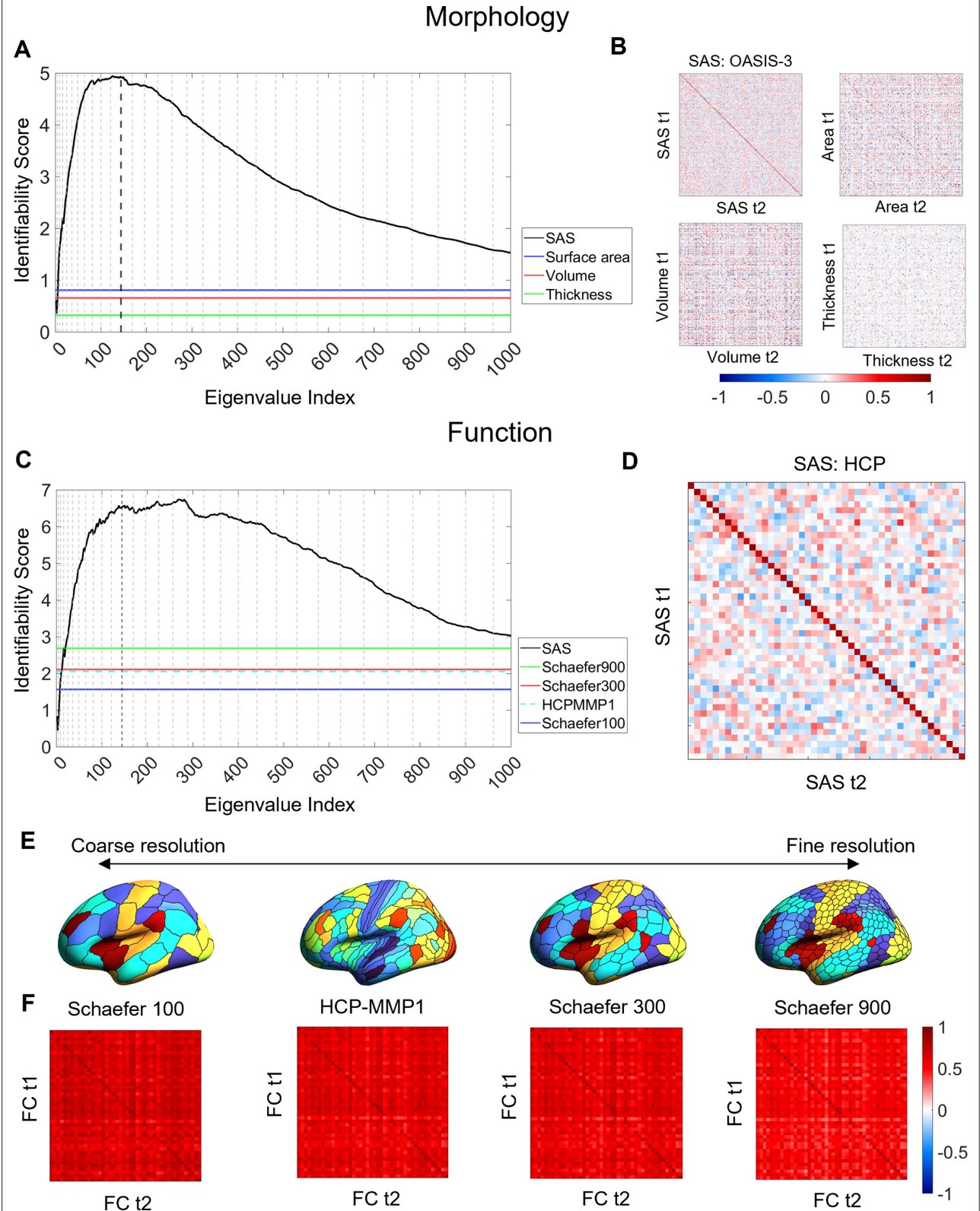

**Figure 3.** Cortical shape asymmetries are more identifiable than size-related descriptors or functional connectivity. (**A**) Identifiability scores for the shape asymmetry signature (SAS) are higher than those obtained for asymmetries based on cortical surface area (identifiability score = 0.81), volume (identifiability score = 0.66), and thickness (identifiability score = 0.33) for the OASIS-3 dataset (n = 232; see 'Materials and methods'). (**B**) Matrices of the Pearson correlation coefficients for SAS and size-based morphological asymmetries from MRI scans taken at different time points (t1 and t2) of

*Figure 3 continued on next page*

*Figure 3 continued*

the OASIS-3 subjects. (**C**) SAS identifiability is higher than the identifiability based on functional connectivity, assessed with parcellations at different resolution scales in the HCP test–retest dataset (n = 44). (**D**) Matrix of the Pearson correlation coefficients for SAS of the HCP test-retest subjects. (**E**) Four resolution scales of parcellations used in the functional connectivity analysis (shown on an inflated fsaverage surface in FreeSurfer). (**F**) Matrices of the Pearson correlation coefficients for functional connectivity using the Schaefer 100 (identifiability score = 1.57), HCP-MMP1 (identifiability score = 2.06), Schaefer 300 (identifiability score = 2.11), and Schaefer 900 (identifiability score = 2.69) parcellations.

The online version of this article includes the following figure supplement(s) for figure 3:

**Figure supplement 1.** Comparing identifiability scores between the shape asymmetry signature (SAS) with either native eigenvalues or volume-normalized eigenvalues.

We used data from 138 monozygotic (MZ) twin pairs, 79 dizygotic (DZ) twin pairs, and 160 of their non-twin siblings drawn from the HCP dataset (*Van Essen et al., 2013*; see 'Materials and methods' for details). Unihemispheric shape descriptors demonstrated strong heritability at very coarse spatial scales and moderate heritability at slightly finer scales. For instance, the heritability of the first eigen-group (second to fourth eigenvalues) of both hemispheres ranged between $0.52 < h^2 < 0.69$ (all $P_{FDR} < 0.05$; *Figure 6A and B*). These eigenvalues are related to shape variations on the coarsest scale that does not include any sulcal or gyral features (the corresponding wavelength is approximately 170 mm). Beyond the second eigen-group, heritability estimates dropped to below 0.5 ($P_{FDR} < 0.05$ for most eigenvalues), and beyond the fourth eigen-group they dropped below 0.3. Most eigenvalues with statistically significant heritability estimates were confined to the first six eigen-groups, which correspond to wavelengths greater than or equal to approximately 65 mm (*Figure 6A and B*, insets). These results indicate that genetic influences on the shape of each cortical hemisphere are expressed over very coarse scales at which only primary cortical folds such as the Sylvian and central sulci are

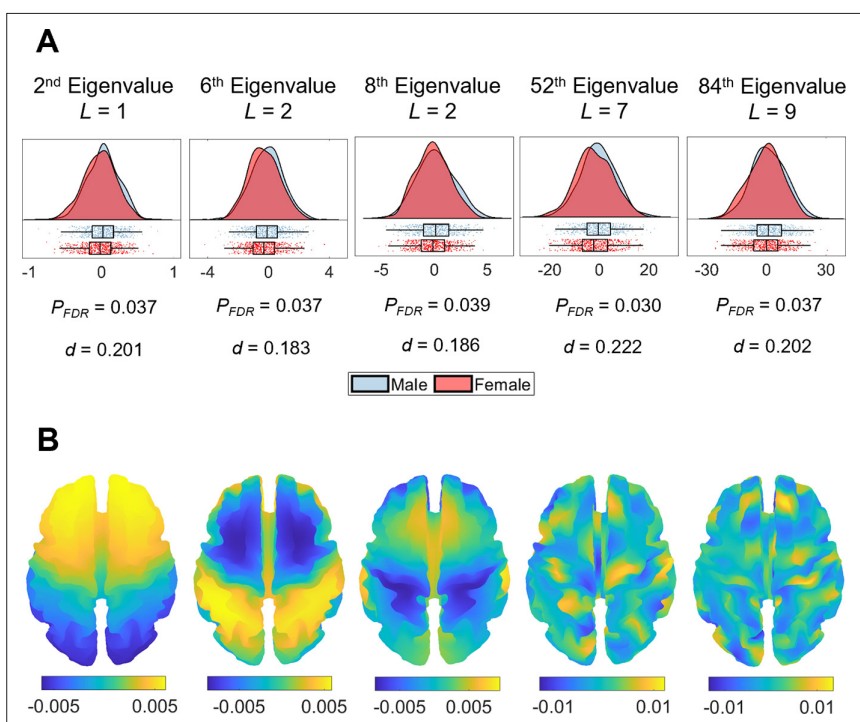

**Figure 4.** Sex differences in eigenvalue asymmetries. (**A**) Smoothed distributions and boxplots with mean and interquartile range (*Allen et al., 2019*) of the eigenvalues among males (n = 504) and females (n = 602). Under these five spatial scales, female brains show a greater rightward asymmetry than males. The p-values are false discovery rate (FDR)-corrected values of the correlation between sex and shape asymmetry signature (SAS), obtained via a general linear model (GLM). The *d* values are effect sizes (Cohen's *d*). *L* denotes eigen-group. (**B**) The corresponding eigenfunction of each eigenvalue in panel (**A**) that shows the gradients of spatial variation on a population-based template.

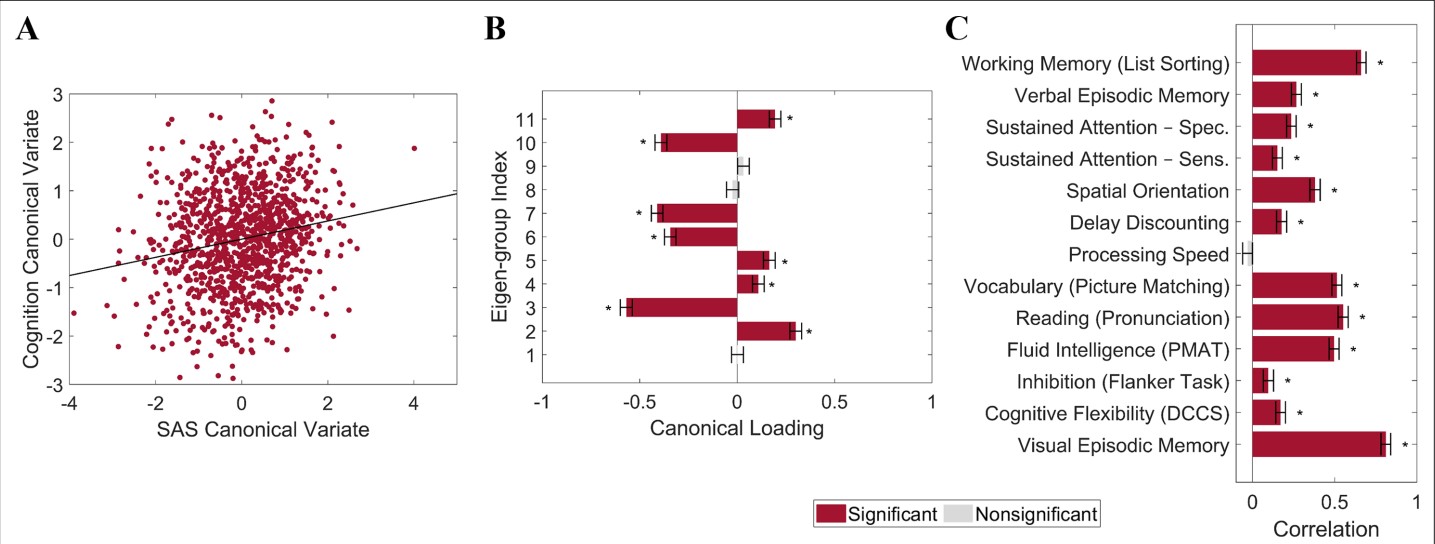

**Figure 5.** Individual differences in cortical shape asymmetry correlate with general cognitive ability. (**A**) Scatterplot of the association between the cognitive and shape asymmetry signature (SAS) canonical variates with the corresponding least-squares regression line in black. (**B**) Canonical variate loadings of each eigen-group. (**C**) Correlations between the original cognitive measures and the cognitive canonical variate. Error bars show ±2 bootstrapped standard errors (SE). Asterisks denote bootstrapped $P_{FDR} < 0.05$. The data in this figure are from the HCP dataset (n = 1106).

apparent. Estimates of common environmental influence on both hemispheres were uniformly low across the 2$^{nd}$ to 144$^{th}$ eigenvalues (range 0–0.20).

In contrast to unihemispheric shape variations, all the heritability estimates of the SAS were low (<0.28; *Figure 6C*), with only four eigenvalues (2, 3, 16, and 28) showing statistically significant heritability after FDR correction ($P_{FDR}$ = 0.004–0.022) and no heritability estimates exceeding 0.30. Thus, at any given scale, genes account for less than one-third of the phenotypic variance in the SAS. These four eigenvalues are confined to the first five eigen-groups, with corresponding wavelengths greater than or equal to approximately 75 mm (*Figure 6C*, inset). Estimates of common environmental influences were uniformly low (range 0–0.14), whereas unique (subject-specific) environmental influences on the SAS were consistently high across the full range of eigenvalues considered, ranging between 0.72 and 1.00 (*Figure 6D*).

Notably, heritability estimates for non-surface area normalized eigenvalues of individual hemispheres, which capture variations in both shape and size, were uniformly high across all scales, and the scale-specific effects were eliminated (*Figure 6—figure supplement 1*), indicating that variations in cortical size are under greater genetic influence than cortical shape. These results underscore the importance of controlling for size-related variations in shape analyses.

## Discussion

Asymmetries in brain anatomy are widely viewed as a critical characteristic for understanding brain function. Here, we employed a multiscale approach to quantify individualized shape asymmetries of the human cerebral cortex. We found that cortical shape asymmetries were highly personalized and robust, with shape asymmetries at coarse spatial scales being the most discriminative among individuals, showing differences between males and females, and correlating with individual differences in cognition. Heritability estimates of shape descriptors in individual hemispheres were high at very coarse scales but declined to moderate values at finer scales. By contrast, the heritability of cortical shape asymmetry was low at all scales, with such asymmetries being predominantly influenced by individual-specific environmental factors.

### Identifiability of cortical shape asymmetry is maximal at coarse scales

Cortical asymmetries have traditionally been investigated at fine-scale, voxel, or vertex-level resolutions (*Cai et al., 2015*; *Takao et al., 2011*; *Kurth et al., 2018*; *Good et al., 2001*; *Maingault et al.,*

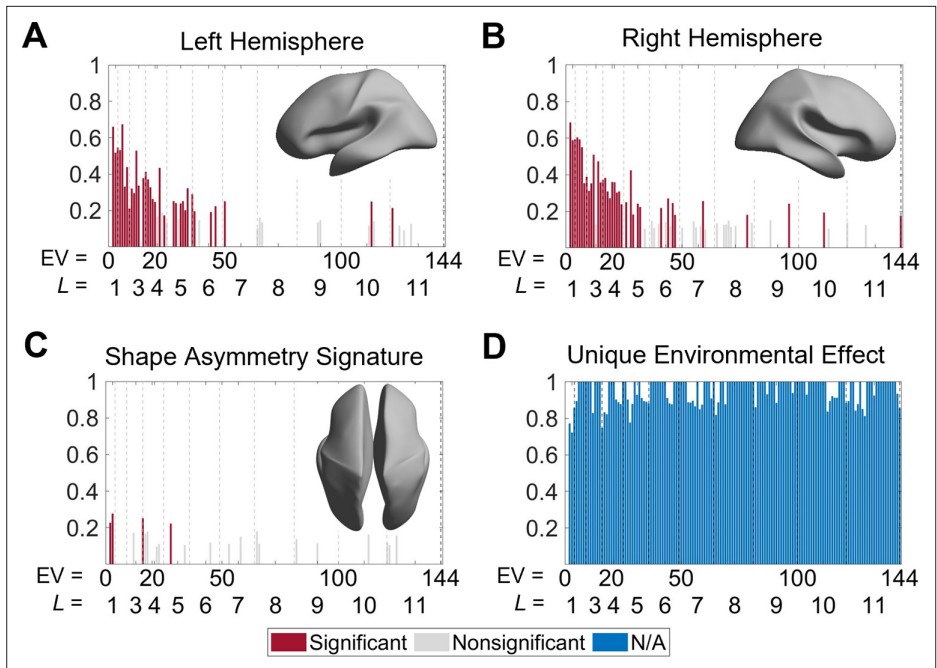

**Figure 6.** Heritability of cortical shape. (**A**, **B**) Heritability of the eigenvalues of the left (**A**) and right (**B**) hemispheres. The insets show the corresponding spatial scales by reconstructing the surfaces using the first six eigen-groups. (**C**) Heritability of the shape asymmetry signature (SAS). The inset shows the corresponding spatial scale with some level of genetic influence, obtained by reconstructing the surface using the first five eigen-groups. (**D**) Unique environmental influences to the SAS at each eigenvalue. Statistical significance is evaluated after false discovery rate (FDR) correction. Note that significance is not estimated for unique environmental effects as this represents the reference model against which other genetically informed models are compared. We use 79 same-sex dizygotic (DZ) twin pairs, 138 monozygotic (MZ) twin pairs, and 160 of their non-twin siblings.

The online version of this article includes the following figure supplement(s) for figure 6:

**Figure supplement 1.** Heritability of cortical shape with volume normalization but without normalizing the surface area.

**Figure supplement 2.** Heritability estimates of regional volumes of individual hemispheres across four parcellation resolutions: Schaefer 100, Schaefer 300, HCP-MMP1, and Schaefer 900 (top to bottom panels).

---

*2016*; *Kruggel and Solodkin, 2020*; *Kurth et al., 2015*). These approaches may ignore meaningful effects (i.e., properties that are individually unique and correlated with cognition) at coarser spatial scales. Our SAS quantifies these underlying variations across the whole brain and along a spectrum of spatial scales. Our approach is akin to studying seismic waves of earthquakes with different wave frequencies at the global tectonic scale, instead of focusing on a particular city. The ability to assess shape along a spectrum of spatial scales is important since brain asymmetry is a multidimensional and multivariate phenotype (*Corballis and Häberling, 2017*; *Kong et al., 2022*; *Kruggel and Solodkin, 2020*).

Few studies have assessed individual variations in shape at coarse scales. *Neubauer et al., 2020* found that individual-specific asymmetry in endocranial shape is reliable across two time points. The endocranial shape is the imprint of the cortical surface shape but contains only very coarse shape information (*Neubauer et al., 2020*). Moreover, levels of brain torque (both horizontal and vertical) are robust across time (*Kong et al., 2021*). *Wachinger et al., 2015* used shape descriptors at coarse scales derived from the eigenvalues of the LBO for all brain structures to achieve accurate subject identification. Taken together with our findings, these results indicate that coarse features of cortical shape are highly personalized and unique to individuals.

It is perhaps surprising that individual differences in cortical shape are most strongly expressed at coarse scales, given the known variability of fine-grained anatomical features such as the presence and trajectories of tertiary sulci. It is possible that local changes in gray matter volume affect fine-scale geometry in such a way that it carries less identifying information, or that such fine scales

carry too much measurement noise to be used for the purpose of identification. Traditional analysis methods use smoothing to address the issue of image noise (*Kurth et al., 2015*), but smoothing can also suppress actual variations at fine scales. Our multiscale approach affords a more comprehensive characterization of shape variations across multiple spatial scales. An important avenue of future work will involve investigating the functional consequences of these pronounced individual differences.

## Cortical shape, rather than shape asymmetry, is heritable

Genetic effects on cortical thickness and surface area are generally bilateral (*Kong et al., 2018*; *Chen et al., 2011*; *Chen et al., 2012*; *Chen et al., 2013*), resulting in few lateralized effects (*Kong et al., 2018*; *Eyler et al., 2014*). Accordingly, it has been postulated that individual-specific asymmetries may be largely determined by environmental factors (*Gómez-Robles et al., 2016*; *Sherwood and Gómez-Robles, 2017*; *Francks, 2015*; *de Kovel et al., 2018*; *Graham and Özener, 2016*). In line with this hypothesis, we found that individualized cortical shape asymmetries were associated with low heritability and were driven mainly by unique environmental effects. These environmental effects are captured by the *E* parameter of the *ACTE* heritability model that we used in our analysis. This parameter also includes the contributions of measurement error. However, our effects cannot be explained by the SAS being a noisier descriptor of morphology as it showed the highest identifiability (*Figure 2A-C*). A noisy measure will not be able to discriminate between individuals in this way. Thus, taking the findings of our identifiability and heritability analyses together, we can conclude that individual differences in SAS scores are primarily driven by unique environmental influences rather than measurement error.

Previous studies have found some evidence of environmental influences on brain asymmetry (*Güntürkün et al., 2020*; *Güntürkün and Ocklenburg, 2017*; *Esteves et al., 2021*; *Felton et al., 2017*). Early in the intrauterine environment, fetal posture and light may influence brain asymmetry (*Güntürkün et al., 2020*; *Güntürkün and Ocklenburg, 2017*; *Esteves et al., 2021*), and during postnatal maturation, language learning has been linked to specific asymmetry features. For example, bilinguals have stronger rightward asymmetry of cortical thickness of the anterior cingulate than monolinguals (*Felton et al., 2017*). However, the mechanisms of how environmental effects shape brain asymmetry are largely unknown, and epigenetics may also play a role (*Güntürkün et al., 2020*; *Güntürkün and Ocklenburg, 2017*).

In contrast to shape asymmetries, the shape of individual hemispheres showed greater heritability at coarse scales, consistent with results from previous studies on other morphological measurements (*Kong et al., 2018*; *Sha et al., 2021*; *Kruggel and Solodkin, 2020*). The scales at which genetic effects on unihemispheric shape were observed captured variations in primary sulci, consistent with evidence that the primary folds, which develop early in life, are less variable across individuals and under greater genetic control than other folds (i.e., secondary and tertiary folds) (*Kruggel and Solodkin, 2020*; *Ronan and Fletcher, 2015*; *Kruggel, 2018*). Previous studies have found that genetic influences on the cerebral thickness, geodesic depth, and surface curvature generally occur along the posterior–anterior and inferior–superior axes (*Kruggel and Solodkin, 2020*; *Valk et al., 2020*). These two axes correspond to the second and third eigenvalues of the LBO, which also showed strong heritability in the shapes of both hemispheres in our results. In addition to these two axes, we found strong heritability at very coarse scales in other directions that have not been described in previous studies. Our approach thus identifies dominant spatial scales and gradients of heritability in shape.

## Shape asymmetries, sex, and handedness

Using our multiscale approach, we did not find a relationship between shape asymmetry and handedness, consistent with numerous studies showing that handedness is unrelated to anatomical brain asymmetry in cortical thickness, volume, surface area, shape, and voxel-based morphometric (VBM) analysis (*Kong et al., 2018*; *Plessen et al., 2014*; *Núñez et al., 2018*; *Wachinger et al., 2015*; *Narr et al., 2007*; *Good et al., 2001*; *Maingault et al., 2016*).

Numerous studies, focusing primarily on size-related descriptions such as gray matter volume and cortical thickness, have found that female brains are more symmetric than male brains (*Zhao et al., 2021*; *Kong et al., 2018*; *Núñez et al., 2018*; *Wachinger et al., 2015*; *Kovalev et al., 2003*; *Guadalupe et al., 2015*). Our analysis reveals that, although the overall level of shape asymmetry did not differ between male and female brains, female brains displayed a greater rightward shape asymmetry

than male brains at certain coarse spatial scales, such as along the anterior–posterior axis. The mechanisms giving rise to these scale-specific sex differences require further investigation.

### Shape asymmetries are correlated with general cognitive performance

We found that individual differences in cortical shape asymmetry are correlated with cognitive performance in a scale-specific way. Specifically, we found that a greater leftward asymmetry across a wide range of spatial scales, corresponding to wavelengths of approximately 37, 75, 95, and 170 mm, and greater rightward asymmetry at wavelengths of approximately 40, 55, 65, and 120 mm, are associated with better performance across nearly all cognitive measures considered. Previous studies have found that asymmetries in cortical thickness and surface area are negatively correlated with cognition (*Nadig et al., 2021*; *Yeo et al., 2016*), but these studies only measured the level of asymmetry and did not consider the direction (i.e., leftward or rightward) of the asymmetry. The scale-specificity of the associations that we find underscores the importance of viewing brain asymmetry as a multiscale rather than a unidimensional trait.

The magnitudes of the associations are modest, but they are consistent with effect sizes reported in past research (*Nadig et al., 2021*; *Yeo et al., 2016*). These modest correlations with cognition may reflect a robustness of cognitive abilities to stochastic perturbations of brain morphology, given that our heritability analysis revealed a dominant effect of unique environmental factors in driving individual differences in cortical shape asymmetries.

### Conclusion

We developed a multiscale approach and found that cortical shape asymmetries are robust and personalized neuroanatomical phenotypes, especially at coarse spatial scales. Some of these coarse scales are more strongly rightward asymmetric in females compared to males. The cortical shape asymmetries also show scale-dependent associations with cognition. Finally, individual-specific cortical shape asymmetries are driven mainly by subject-specific environmental influences rather than by genetics, contrasting with the shape of individual hemispheres, which shows strong heritability at coarse scales.

## Materials and methods
### Neuroimaging data

We used healthy subject data from three open-source neuroimaging datasets: the latest OASIS-3 (*LaMontagne et al., 2019*), the HCP (*Van Essen et al., 2013*), and the ADNI (https://ida.loni.usc.edu/) to develop and test our new asymmetry shape measure – the SAS (see below for details). To test for relationships of sex, handedness, and heritability, we restricted our analysis to the HCP dataset, which provides twin and non-twin sibling information and handedness measurement as a continuous variable, as the sample sizes of the left-handers in the other two datasets are too small (n = 15 in the ADNI data; n = 18 in the OASIS-3 data).

### OASIS-3

We used 239 healthy participants with at least two longitudinal MRI sessions using 3T scanners from the latest release of the OASIS-3 (*LaMontagne et al., 2019*). We excluded six subjects whose SAS was an outlier in at least one of those sessions due to poor image quality and major errors in image segmentation. These subjects had more than two eigenvalues of the first 200 eigenvalues that departed from the population mean values by more than four standard deviations. The remaining 233 subjects (99 males; 134 females) were aged from 42 to 86 (mean = 66.03; standard deviation = 8.81) when they entered the study. We also repeated the analyses using all the subjects including the outliers, and the resulting number of eigenvalues with peak identifiability was identical to the initial analysis that excluded the outliers. For comparing the identifiability of the SAS and the asymmetry from traditional measurements (volume, cortical thickness, and surface area), we further excluded one subject because some of this subject's files were corrupted and could not be segmented. For subjects with more than two MRI sessions (n = 115), our main analysis used the initial session as the time 1 (t1) session and the session closest in time to the initial session as the time 2 (t2) session. The intervals between these two sessions were one to 3151 days (mean = 2.95 years; standard deviation = 1.67 years). To ensure the robustness of our methods, we used sessions with the longest intersession interval (mean interval

of 6.24 years; standard deviation of 1.88 years) to reanalyze the subject identifiability. These healthy participants had no history of neurological or psychiatric diseases. We also excluded subjects with a Mini-Mental State Examination (MMSE) score equal to or lower than 26 as this indicates that a subject is at risk of being diagnosed with dementia (*O'Bryant et al., 2008*).

OASIS-3 (*LaMontagne et al., 2019*) provides surface meshes based on the T1-weighted MRI images created by FreeSurfer version 5.3 with the cross-sectional pipeline (i.e., to treat the T1 and T2 sessions independently; *Fischl et al., 2002*), including the FreeSurfer patch (10 December 2012) and the HCP patch (http://surfer.nmr.mgh.harvard.edu/pub/dist/freesurfer/5.3.0-HCP; *LaMontagne et al., 2019*). A trained lab member of the OASIS project reviewed the image segmentation, and for the images that failed the quality control, TkMedit (http://freesurfer.net/fswiki/TkMedit), a FreeSurfer toolbox, was used to revise the images and rerun the FreeSurfer pipeline (*LaMontagne et al., 2019*). After the re-segmentation, the images were excluded if they still failed a quality control process (*LaMontagne et al., 2019*). The details of the OASIS-3 dataset can be found in *LaMontagne et al., 2019* and the OASIS website (https://www.oasis-brains.org/). We used the actual output files provided by the OASIS-3 without any further corrections.

## HCP

We used participants from the HCP (*Van Essen et al., 2013*) s1200 release (https://www.humanconnectome.org/), which includes 1113 subjects with T1-weighted MRI. All subjects of the s1200 release were healthy young adults (aged 22–35, mean = 28.80, standard deviation = 3.70). The structural images (T1-weighted and T2-weighted scans) of the HCP have a high isotropic resolution (0.7 mm; see *Van Essen et al., 2013* for details), and all images underwent the HCP-specific minimal preprocessing pipeline (*Glasser et al., 2013*). We used native surface meshes created by the FreeSurfer (version 5.3)-HCP pipeline (*Fischl et al., 2002*; *Glasser et al., 2013*; *Jenkinson et al., 2012*; *Jenkinson et al., 2002*) from T1-weighted MRI images using 3T scanners. For subject identification, we employed the test–retest subsample, which consists of 45 healthy subjects (13 males, 32 females) aged from 22 to 35 (mean = 30.29; standard deviation = 3.34), including 17 pairs of MZ twins. The intervals between the test session (the t1 session in our analysis) and the retest session (t2) were between about 1 and 11 months (mean interval of 4.7 months). To compare the identifiability of the SAS and the resting-state functional connectivity, we further excluded one subject without REST1 data in one session.

For analyzing the relationships between SAS and sex as well as handedness, we excluded three subjects with unclear zygosity and four subjects with outlying SAS values (using the same criteria as used in the OASIS-3) from the s1200 release subjects, and GLM of sex and handedness effects were applied to cross-sectional data of these remaining 1106 subjects (504 males; 602 females). We further excluded 12 subjects who did not have all 13 cognitive measures analyzed in our CCA (detailed below). Among the s1200 release subjects were 79 same-sex DZ twin pairs and 138 MZ twin pairs; 160 of these twin pairs have non-twin sibling imaging data. For twin pairs with more than one non-twin sibling, we selected one sibling at random (*Arnatkeviciute et al., 2021*). We used the resulting twin and non-twin siblings data for the heritability analysis.

## ADNI

The ADNI database (adni.loni.usc.edu) was launched in 2003 as a public–private partnership, led by principal investigator Michael W. Weiner, MD. The primary goal of ADNI has been to test whether serial MRI, positron emission tomography (PET), other biological markers, and clinical and neuropsychological assessment can be combined to measure the progression of mild cognitive impairment (MCI) and early Alzheimer's disease (AD).

Participants in the ADNI sample completed multiple MRI sessions, but the number of sessions was not consistent across subjects. We used 208 healthy control subjects from the ADNI 1 who had both the baseline MRI session (the t1 session) and a follow-up MRI session 6 months later (the t2 session). These subjects comprised 109 males and 99 females aged 60–90 (mean = 76.21; standard deviation = 5.10) upon study entry. Of these 208 subjects, 135 subjects also had an MRI session 3 years later from the initial session. To evaluate the stability of our methods, we reanalyzed these 135 subjects using data from the 3-year follow-up as the t2 session. The preprocessing procedure included gradwarping, B1 correction, and/or N3 scaling. We used the ADNI-provided surface meshes generated by the cross-sectional FreeSurfer (version 4.3) from T1-weighted MRI image. Detailed descriptions of image

acquisition, quality control, and preprocessing are described at http://adni.loni.usc.edu/methods/mri-tool/mri-analysis/ and *Jack et al., 2008*.

## Spectral shape analysis

We utilized the eigenvalues of the LBO applied to cortical surface mesh models generated with Free-Surfer (*Fischl et al., 2002*). The eigendecomposition of each individual's cortical surface was estimated using the Shape-DNA software (*Wachinger et al., 2015*; *Reuter et al., 2009*; *Reuter et al., 2006*), which provides algorithms that extract and optimize the eigenvalues and eigenfunctions from the LBO based on the intrinsic shape of an object (*Reuter et al., 2009*; *Reuter et al., 2006*). The Shape-DNA software (*Reuter et al., 2009*; *Reuter et al., 2006*) uses the cubic finite element method to solve the Helmholtz equation (*Equation 1*), also known as the Laplacian eigenvalue problem:

$$\Delta f = -\lambda f \tag{1}$$

where $\Delta$ is the LBO, and $f$ is the eigenfunction with corresponding eigenvalue $\lambda$. The eigenvalues of the Helmholtz equation are a sequence ranging from zero to infinity, that is, $0 \leq \lambda^1 \leq \lambda^2 \leq \ldots < \infty$, and changes in shape result in changes in the eigenvalue spectrum (*Reuter et al., 2006*).

Spectral shape analysis via LBO is a departure from traditional morphological analyses that focus on either specific locations (i.e., regions defined by a cortical atlas) or global differences (such as total hemispheric volume). Spectral shape analysis focuses instead on differences in the spatial scales of variation. The decomposed spatial scales can be linearly combined to reconstruct the surface via the eigenfunctions and their corresponding coefficients (the contribution of each set of eigenfunctions to the original surface; see *Figure 2G* for examples of reconstructed surfaces).

Importantly, Shape-DNA achieves better results for retrieving object shapes than numerous cutting-edge shape-retrieval methods (*Lian et al., 2013*). Shape-DNA compresses the cortical-surface geometry from around 5 mb into only less than 3 kb, making it computationally efficient for further analysis (*Wachinger et al., 2015*). The code for calculating Shape-DNA is written in Python and is freely available (http://reuter.mit.edu/software/shapedna/). We applied the Shape-DNA code to the data and analyzed the resulting eigenvalues using MATLAB.

## Eigenvalue normalization

To account for differences in brain sizes among participants, the eigenvalue spectra from Shape-DNA should be normalized (*Reuter et al., 2009*). Previous studies *Wachinger et al., 2015*; *Wachinger et al., 2016*; *Wachinger et al., 2018* have applied volume normalization to normalize the eigenvalue spectrum to unit volume via the following equation (*Wachinger et al., 2015*; *Wachinger et al., 2016*):

$$\lambda' = v^{2/D} \lambda \tag{2}$$

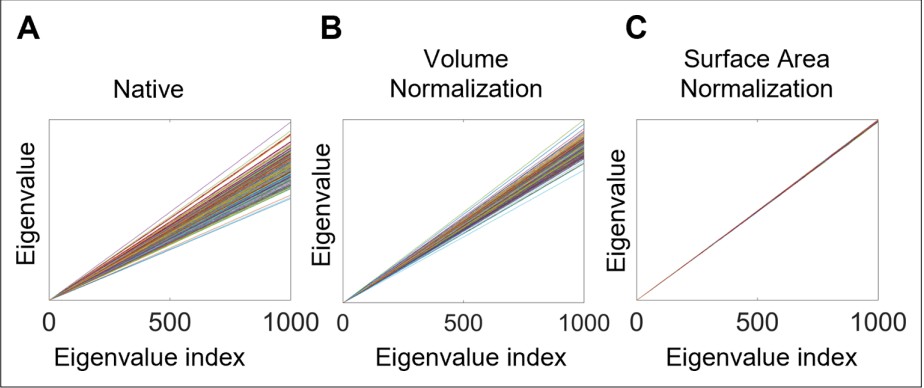

**Figure 7.** Eigenvalue spectra with and without normalization. (**A**) Native eigenvalue spectra. (**B**) Eigenvalue spectra with volume normalization. (**C**) Eigenvalue spectra with surface area normalization. All of these results are from the left white surfaces of 233 subjects from the OASIS-3 data. Each line represents a subject. The slopes of the spectra in (**A**) and (**B**) differ among subjects, whereas those in (**C**) are almost identical.

where $v$ is the Riemannian volume of the manifold, $\lambda$ is the original eigenvalue spectrum ($\lambda = [\lambda^1, \lambda^2, \ldots]$), and $\lambda$ is the volume normalized eigenvalue spectrum. Although this approach has been used in the literature, it is still unable to isolate shape properties as it does not control the effect of different surface areas among objects. For example, in *Figure 7*, each line is the eigenvalue spectrum for the cortical surface of one subject, and these eigenvalue spectra are straight lines (although they are not straight lines if we zoom in these figures) increasing along with the indices: each eigen-spectrum line has its own slope. Specifically, slopes of the native eigenvalue spectra from each subject are different (*Figure 7A*) and related to the volume of the manifold. Even though volume normalization decreases the differences in the slopes of the eigenvalue spectra, the slopes remain quite different (*Figure 7B*) and are driven by differences in surface area (*Reuter et al., 2009*). More specifically,

$$\lambda n \sim \frac{4\pi n}{area(M)} \tag{3}$$

where $\lambda$ is the eigenvalue and $n$ is the eigenvalue index. Hence, an appropriate surface area-based normalization is essential to isolate the effects of shape that are distinct from size, particularly given the evidence that the right hemisphere tends to have a greater cortical surface area than the left hemisphere (*Kong et al., 2018*). Without surface area normalization, differences between the hemispheres may be primarily driven by differences in the surface area of the two hemispheres.

To perform surface area normalization, we obtained the unit surface area by dividing the vertex coordinates on each axis by the square root of the total surface area (*Equation 4*).

$$Vx' = \frac{Vx}{\sqrt{area(M)}}; Vy' = \frac{Vy}{\sqrt{area(M)}}; Vz' = \frac{Vz}{\sqrt{area(M)}} \tag{4}$$

where Vx, Vy, Vz are the coordinates of all vertices on the X-axis, Y-axis, and Z-axis, respectively; area ($M$) is the surface area of object $M$; Vx', Vy', Vz' are the coordinates of transformed vertices on the X-axis, Y-axis, and Z-axis, respectively. Surface area normalization is stricter than volume normalization for spectral alignment, and the eigenvalue spectra with surface area normalization have a nearly identical slope (*Reuter et al., 2009*; *Figure 7C*).

## The shape asymmetry signature

The LBO eigenvalues measure the intrinsic geometry of an object and are isometry invariant. Hence, the eigenvalue spectra are identical regardless of object positions, rotations, and mirroring (i.e., perfect projection from the brain structure of the right hemisphere to the left does not change the eigenvalue spectrum) (*Wachinger et al., 2015*; *Reuter et al., 2006*). Therefore, brain asymmetry can be calculated directly from the eigenvalue spectra of the two hemispheres (*Wachinger et al., 2015*; *Wachinger et al., 2016*) without image registration or smoothing (*Wachinger et al., 2015*; *Reuter et al., 2006*). In this study, after calculating the eigenvalues with surface area normalization, we subtracted the eigenvalue spectra of the right hemisphere from those of the left hemisphere in the same subject at each spatial scale (each eigenvalue index) to define the SAS. Formally, the SAS for subject $i$ is given by

$$\Lambda^i = \lambda_L^i - \lambda_R^i \tag{5}$$

where $\lambda$ is the eigenvalue spectrum $\lambda = (\lambda^1, \lambda^2, \ldots, \lambda^n)$ from the left ($L$) and right ($R$) hemispheres, each of which represents a certain spatial scale. There are other possible asymmetry indices (*Kong et al., 2018*; *Moodie et al., 2020*), but those indices are not appropriate for a surface area-normalized eigenvalue analysis, as our normalization already accounts for size effects.

*Moodie et al., 2020* proposed subtracting the mean of the asymmetry values across subjects from the individual asymmetry values to represent the asymmetry. We tested this approach with our method, and the results were generally the same, as the eigenvalues were normalized before calculating the asymmetry. For simplicity, we defined the SAS using *Equation 5* to represent the individual-specific asymmetry.

To further check the possible influence of image quality on the SAS, we first took the mean of the Euler number of the left and right hemispheres using FreeSurfer, which is widely used as an index of image quality (*Morgan et al., 2019*; *Kaufmann et al., 2019*; *Rosen et al., 2018*), and then calculated the Pearson's correlation between the mean Euler number and the SAS across the first 200

eigenvalues. For the HCP s1200 dataset, the correlations were all below 0.07 ($P_{FDR} > 0.05$). For the OASIS-3, the correlations were all below 0.18 ($P_{FDR} > 0.05$) at either time 1 or time 2 MRI session. These results indicate that image quality does not strongly influence the SAS, which is in line with past findings that the eigenvalues and eigenfunctions of the LBO are robust to image noise (*Reuter, 2010*).

## Subject identification

Our first aim was to validate the SAS as a useful and robust measure of individual-specific asymmetry. We, therefore, evaluated the extent to which the SAS of each individual subject measured at time 1 (t1) could be used to identify the same person at time 2 (t2) in the longitudinal data, akin to a neuro-morphological fingerprint. The t1 – t2 Pearson correlations were then estimated between all pairs of $N$ individuals, resulting in an $N \times N$ correlation matrix.

*Amico and Goñi, 2018* defined identifiability as the difference between the mean of within-subject correlations (diagonal elements of the Pearson correlation matrix in *Figure 1C*) and the mean of between-subject correlations (off-diagonal elements of the Pearson correlation matrix in *Figure 1C*). This approach allows for a more quantitative and finer-grained comparison of the identifiability of different metrics compared to other approaches that just rely on binary identification accuracy (e.g., *Finn et al., 2015*; *Amico and Goñi, 2018*; *Mansour L et al., 2021*). However, this approach does not take into account the variance of the observations. To examine the within- and between- subject similarities, we utilized the Glass's Δ, which is the standardized difference between the mean values of two categories of observations, normalized by the standard deviation of a control group (*Glass et al., 1981*), which is the between-subject group in our case. Glass's Δ has been recommended when the standard deviations of the two groups are substantially different (*Glass et al., 1981*; *Lakens, 2013*), which is the case for the between- and within-subject groups. Thus, our identifiability score was given by

$$Identifiability\ score = \frac{mean(r_{ii}) - mean(r_{ij})}{SD(r_{ij})} \tag{6}$$

where *SD* is the standard deviation. Higher scores indicate a greater capacity to discriminate between individuals. We also tested the pooled standard deviation of the two groups (*Mansour L et al., 2021*), as in the estimation of Cohen's *d*, and the results were generally consistent to those using the Glass's Δ.

We also evaluated the identifiability performance of the SAS with respect to unihemispheric descriptors of either combining size and shape or shape alone: namely, the eigenvalues (native, volume-normalized, or surface area-normalized) from the same hemispheres between time 1 and time 2 follow-up; concatenating eigenvalues of both left and right hemispheres between time 1 and time 2; and identifying the shape of one hemisphere from the shape of the other hemisphere both at time 1 or both at time 2. Finally, we compared the identifiability score of the SAS to the asymmetry based on commonly used size-related measures (i.e., volume, cortical thickness, and surface area), and resting-state functional connectivity.

## Identifying spatial scales for optimum subject identifiability

Given a surface of $N$ vertices, spectral shape analysis yields up to $N$ eigenvalues, raising the question of how many eigenvalues constitute a sufficient description of cortical shape. Is a full representation of the entire surface necessary for optimal subject identifiability, or can this be achieved using a more compact set of eigenvalues? If so, the specific number of eigenvalues required would define the relevant spatial scale of shape differences that characterize the individual-specific asymmetry at which individual differences are most prevalent.

To address this question, we decomposed the cortical surface and use an increasing number of eigenvalues, from the first two eigenvalues ($\lambda^1$, $\lambda^2$) to the first 1000 eigenvalues ($\lambda^1$, $\lambda^2, \lambda^3, ..., \lambda^{1000}$), each time computing the SAS and evaluating subject identifiability. For example, we first quantified the shape of cortical surface using only $\lambda^1$ and $\lambda^2$, thus capturing the coarsest scales of cortical shape. We then quantified the surface using $\lambda^1$ through $\lambda^3$, then $\lambda^1$ through $\lambda^4$, and so on. If there is a specific spatial scale that is optimal for this subject identifiability, we expect to see a peak in the identifiability score as a function of the truncation number, $k$. This peak not only defines the spatial scale at which individual variability, and thus individual-specific asymmetry, is most strongly expressed, but

it also identifies a meaningful point at which to define a compressed summary of individual-specific asymmetry using the eigenvalue spectrum.

## Cortical shape harmonics

The cerebral cortex is topologically equivalent to a sphere. Solving the Helmholtz equation for a sphere yields groups of eigenfunctions with the same eigenvalues and spatial wavelength, progressing along orthogonal axes (*Robinson et al., 2016*). These groups in the solutions to the idealized spherical case are known as the spherical harmonics. The zeroth group ($L = 0$) is comprised of the first eigenvalue; the first group ($L = 1$) is comprised of the second, third, and fourth eigenvalues; the second group ($L = 2$) is comprised of the fifth to ninth eigenvalues, and so on, with 2 ($L + 1$)–1 eigenvalues in the $L$th group. *Robinson et al., 2016* showed that while the eigenvalues between the cortical surface and sphere are different, the spherical grouping provides a rough division of the convoluted cortical surface. This is a useful grouping approach to investigate eigenfunctions and eigenvalues as the constituents of each group have roughly the same spatial wavelength. By averaging over several eigenvalues with similar spatial scales, we can also increase the stability of the truncation number across datasets. For example, the peak SAS identifiability appeared at the first 126 and 122 eigenvalues for the OASIS-3 and ADNI data, respectively, and these eigenvalues are all within the 11h eigen-group ($L = 11$).

To estimate the corresponding wavelength of each eigen-group, we used an approximation of the spatial wavelength in the spherical case:

$$W = \frac{2\pi Rs}{\sqrt{L(L+1)}} \tag{7}$$

where $Rs$ is the equivalent sphere of the original object (for the fsaverage case, $Rs$ is about 67 mm) and $L$ is the index of the eigen-group. We used the population-based template (fsaverage) as an example to show the wavelengths of the first 14 eigen-groups in *Supplementary file 1*.

## Cortical segmentation

We applied the HCP-MMP1 atlas (*Glasser et al., 2016*) to segment cortical regions for accessing size-related morphological asymmetry, functional connectivity, and regional volume heritability. This atlas is based on a surface alignment approach that aligns the images using cortical folding patterns and minimizes the spatial smoothness (*Glasser et al., 2016*; *Coalson et al., 2018*), thus offering more accurate inter-subject registration than volume-based registration (*Glasser et al., 2016*). Moreover, regions in the left and right hemispheres of the HCP-MMP1 atlas are corresponding and thus can be used for accessing cortical asymmetry. In addition to the HCP-MMP1 atlas, we also employed the Schaefer atlas (Schaefer 100, 300, and 900) (*Schaefer et al., 2018*) for constructing functional connectivity (FC) and regional volume heritability. The Schaefer atlas has superior functional homogeneity and has different parcellation scales (*Schaefer et al., 2018*); therefore, it can be used for comparing the identifiability of the FC and estimating regional volume heritability at different scales. Specifically, each hemisphere has 50 regions in the Schaefer 100 atlas, 150 regions in the Schaefer 300 atlas, and 450 regions in the Schaefer 900 atlas (*Schaefer et al., 2018*). However, regions in the left and right hemispheres of the Schaefer atlas are not corresponding; therefore, the atlas cannot be used for assessing brain asymmetry.

## Non-shape descriptors of brain anatomical asymmetry

To compare identifiability scores obtained with SAS to asymmetries using size-related descriptors, including volume, cortical thickness, and surface area, we had to ensure that the asymmetry values were purely from the asymmetry effect and were not affected by the effect of total brain size. A traditional asymmetry index (*Kong et al., 2022*; *Kurth et al., 2018*; *Sha et al., 2021*) is

$$AI^{S,i} = \frac{\left(P_L^{S,i} - P_R^{S,i}\right)}{0.5\left(P_L^{S,i} + P_R^{S,i}\right)} \tag{8}$$

where $AI^{S,i}$ is the asymmetry index for parcellation $S$ of subject $i$. $P_L^{S,i}$ is the value of the morphological measurement from parcellation $S$ from subject $i$'s left hemisphere; and $P_R^{S,i}$ is from the right hemisphere. However, this asymmetry index is unable to entirely eliminate the effect of total brain size.

Here, we use cortical volume as an example. We suppose the total brain volume effect (α) exists, and the effects of each ROI-based volume of the left (*Equation 9*) and right (*Equation 10*) hemispheres are

$$V_L^{F,i} = \delta_L^i V_L^i + \alpha^i \tag{9}$$

$$V_R^{F,i} = \delta_R^i V_R^i + \alpha^i \tag{10}$$

where $V_L^i$ and $V_R^i$ are the volumes of region $i$ in the left and right hemispheres, respectively, $\delta_L^i$ and $\delta_L^i$ are the scaling coefficients, and $\alpha i$ is the effect of total brain volume on region $i$. Thus, $V_L^{F,i}$ and $V_R^{F,i}$ are the overall effects of volume on region $i$. We can apply $V_L^{F,i}$ and $V_R^{F,i}$ to the traditional asymmetry index as in *Equation 8* to get

$$V_{AI}^i = \frac{[(\delta_L^i V_L^{F,i} + \alpha^i) - (\delta_R^i V_R^{F,i} + \alpha^i)]}{0.5[(\delta_L^i V_L^{F,i} + \alpha^i) + (\delta_R^i V_R^{F,i} + \alpha^i)]} \tag{11}$$

By rearranging this equation, we obtain

$$V_{AI}^i = \frac{\delta_L^i V_L^{F,i} - \delta_R^i V_R^{F,i}}{0.5\delta_L^i V_L^{F,i} + 0.5\delta_R^i V_R^{F,i} + \alpha^i} \tag{12}$$

which shows that the total volume effect $\alpha^i$ still remains in the denominator and is not removed by the traditional asymmetry index defined in *Equation 8*.

In this study, we adjusted the asymmetry index for the mean of each morphological measurement, such as the asymmetry of cortical thickness, volume, and surface area. Specifically, we revised the traditional asymmetry index by subtracting the mean value of the measurement across all parcellations of each subject before calculating the asymmetry index defined in *Equation 8*. This revised asymmetry measure $RAI^{S,i}$ is explicitly calculated as

$$RAI^{S,i} = \frac{\left(P_L^{S,i} - M^i\right) - \left(P_R^{S,i} - M^i\right)}{0.5\left[\left(P_L^{S,i} - M^i\right) + \left(P_R^{S,i} - M^i\right)\right]} \tag{13}$$

where $M^i$ is the mean value of the measurement across all regions in parcellation of subject $i$. We note that this is an important point as without this correction, the asymmetry measure is dependent on the mean value.

We employed a multi-modal parcellation, HCP-MMP1 version 1.0 (*Glasser et al., 2016*), on the OASIS-3 subjects. We excluded one subject whose cortical surfaces could not be segmented by the HCP-MMP1 atlas. There are 180 regions in each hemisphere of the HCP-MMP1 atlas, including the hippocampus that was excluded in our analysis. We created one vector per size-related measure that quantified the asymmetry index per subject and then used these asymmetry indices in the subject identifiability analyses.

## Functional connectivity

We used the resting-state FC from the first session (REST1) in the test sample as the first FC time point (t1) and FC from the first session in the retest session as the second FC time point (t2). We utilized the fMRI signals that were preprocessed by the HCP functional and ICA-FIX pipelines (*Glasser et al., 2013*). We did not apply any spatial smoothing on the signals. FC was calculated using the upper triangle entries of the Pearson correlation matrix between nodes from the atlas (*Finn et al., 2015*). To compare the identifiability of the SAS and the FC across different parcellation scales and atlas, we repeated the FC analysis with the Schaefer 100, 300, and 900 atlas (*Schaefer et al., 2018*) and the HCP-MMP1 atlas (*Glasser et al., 2016*) from the subjects in the HCP test–retest subsample (n = 44; we excluded one subject without REST1 data in one session).

## Relationships with sex and handedness

Sex and handedness are two common characteristics that have been widely examined in the asymmetry literature (*Corballis and Häberling, 2017*; *Güntürkün and Ocklenburg, 2017*; *Toga and Thompson, 2003*; *Kong et al., 2018*; *Plessen et al., 2014*; *Kong et al., 2022*; *Deep-Soboslay et al., 2010*; *Núñez et al., 2018*; *Wachinger et al., 2015*; *Narr et al., 2007*; *Steinmetz et al., 1991*; *Good et al., 2001*; *Guadalupe et al., 2014*; *Maingault et al., 2016*; *Kong et al., 2021*). We used a GLM

to analyze relationships between each eigenvalue with sex and handedness on 1106 HCP subjects (see 'HCP' section). The HCP dataset provides the handedness preference measured by the Edinburgh Handedness Inventory (EHI) (*Oldfield, 1971*). EHI is the most widely used handedness inventory (*Vlachos et al., 2013*; *Willems et al., 2014*), with resulting scores range from –100 (complete left-handedness) to 100 (complete right-handedness) (*Oldfield, 1971*). Handedness preference is not a bimodal phenomenon (*Dragovic, 2004*), and cutoff scores to categorize the handedness are still arbitrary. We therefore used the EHI score as a continuous variable in our main analysis, which is a widely used approach (*Maingault et al., 2016*; *Kong et al., 2021*). To further confirm the robustness of the relationship between handedness and the SAS, we tested two thresholds to categorize handedness. First, right-handed (EHI: 71–100), left-handed (EHI: –100 to -71), and ambidextrous (EHI: –70–70) (*Deep-Soboslay et al., 2010*; *Narr et al., 2007*; *Dragovic, 2004*) second, right-handed (EHI: 50–100), left-handed (EHI: –100 to -50), and ambidextrous (EHI: –49–49) (*Vlachos et al., 2013*; *Perlaki et al., 2013*). Regardless of the threshold, the categorized handedness variable was still unrelated to the SAS (2–144 eigenvalues).

## Relationships with cognition

We followed *Kong et al., 2019* and used 13 cognitive measures in the HCP data dictionary that represent a wide range of cognitive functions, namely, (1) Visual Episodic Memory (PicSeq_Unadj); (2) Cognitive Flexibility (CardSort_Unadj); (3) Inhibition (Flanker_Unadj); (4) Fluid Intelligence (PMAT24_A_CR); (5) Reading (ReadEng_Unadj); (6) Vocabulary (PicVocab_Unadj); (7) Processing Speed (ProcSpeed_Unadj); (8) Delay Discounting (DDisc_AUC_40K); (9) Spatial Orientation (VSPLOT_TC); (10) Sustained Attention – Sens (SCPT_SEN); (11) Sustained Attention – Spec (SCPT_SPEC); (12) Verbal Episodic Memory (IWRD_TOT); and (13) Working Memory (ListSort_Unadj). We used PCA to reduce dimensionality and minimize collinearity in the CCA. The first four PCs explained 80% of the variance and were retained for our primary analysis. Similarly, we reduced the dimensionality of the SAS measures and ensured equal representation across different spatial scales by taking the mean of the SAS across each eigen-group (from 1st to 11th groups). These 11 mean SAS values and the first 4 cognitive PCs were then subjected to CCA to identify linear combinations of SAS and cognitive measures that maximally covary with each other (*Smith et al., 2015*). Inference on the resulting canonical variates was performed using a permutation-based procedure (*Winkler et al., 2020*), and robust estimates of canonical loadings were obtained using bootstrapping (*Dong et al., 2020*), as described in the 'Statistical analysis' section.

## Heritability of brain shape

We used MZ and same-sex DZ twin pairs and their non-twin siblings to calculate the heritability of brain shape and cortical volume. For twin pairs with more than one non-twin sibling, we selected one sibling at random. We estimated the heritability of each eigenvalue from individual hemispheres and the SAS. To emphasize the importance of properly controlling surface area, we show the heritability of eigenvalues with volume normalization (but without surface area normalization; *Figure 6—figure supplement 1*). We also calculated the heritability from ROI-based volumes of individual hemispheres (*Figure 6—figure supplement 2*). Regional cortical volumes of individual hemispheres were estimated for each region of the Schaefer 100, 300, and 900 atlas (*Schaefer et al., 2018*), as well as the HCP-MMP1 atlas (*Glasser et al., 2016*).

Under the assumption that MZ twins are genetically identical whereas DZ twins on average share half of their DNA, structural equation modeling (SEM) can be used to decompose the phenotypic variance of a trait into additive genetic (A), common environmental (C), and unique (subject-specific) environmental (E) factors (*Arnatkeviciute et al., 2021*). Twins raised together are likely to share a more common environment compared to their non-twin siblings; therefore, including a set of non-twin siblings into the analysis allows us to additionally separate common environmental contributions into twin-specific (T) and twin non-specific common environmental factors (C). The heritability analyses of brain shape and volume were performed independently using standard SEM implemented in OpenMx software (*Boker et al., 2011*; *Neale et al., 2016*) in R.

For each eigenvalue and parcellated volume, outlying values were first excluded using the boxplot function in R keeping data points (v) in a range Q1−1.5 × IQR < v < Q3 + 1.5 × IQR, where Q1 and Q3 are the first and third quartiles respectively, and IQR is the interquartile range (*Arnatkeviciute et al.,*

*2021*). For each phenotype, we then fitted a set of biometric models – ACTE, ACE, ATE, CTE, TE, CE, E – using age and sex as covariates, where the letters indicate the factors present in the model. The goodness of fit between the models was compared using the Akaike information criterion (AIC) (*Parzen et al., 1998*), and the best-fitting model for each measure was selected based on the lowest AIC value. Consequently, the heritability for each measure was derived from the best-fitting model, corresponding to the best model that balances the ability to explain data with model complexity. To ensure that the general heritability pattern was not dependent on the model selection, we also calculated the heritability estimates from the full ACTE model (without model selection) at each eigenvalue (with surface area normalization) of individual hemispheres as well as the SAS. The heritability estimates were highly correlated with those with model selection (Pearson correlation $r$ = 0.92–0.96).

## Statistical analysis

We applied a permutation test to evaluate the statistical significance of a given identifiability score for a given number of eigenvalues. We randomly shuffled the subject order of the SAS of the t2 session 50,000 times and then compared the original identifiability score with all the permuted peak identifiability score truncated at each iteration independently to calculate the $P$-value. Statistical inference for models evaluating associations between SAS and sex and handedness was also performed using a permutation test with 100,000 iterations by randomly shuffled the subjects' sex and handedness data.

When analyzing associations between the SAS and cognition, we used a recently developed permutation-based procedure for CCA inference (*Winkler et al., 2020*) with 50,000 iterations. The $P$-values of the canonical modes were controlled over family-wise error rate (FWER; FWER corrected $P$-values are denoted $P_{FWER}$), which is more appropriate than the FDR when measuring the significant canonical mode (*Winkler et al., 2020*). The results were consistent when controlling for age and sex as confounding variables. To identify reliable loadings of each SAS eigen-group on the canonical variate, we used bootstrapping with 1000 iterations of the correlation between each SAS eigen-group and the SAS canonical variate. The resulting standard errors were used to estimate z-scores for each loading by dividing the original correlation by the standard errors, and then the z-scores were used to compute two-tailed p-values. We then used FDR ($q$ = 0.05) to correct for multiple comparisons of $P$-values of all the eigen-groups. We also used the same approach to identify reliable correlations of cognitive measures on the corresponding canonical variate. Due to the family structure of the HCP data, we kept the subjects' family structures intact when shuffling or bootstrapping the subjects using the Permutation Analysis of Linear Models (PALM) software package (*Winkler et al., 2016*; *Winkler et al., 2015*).

The statistical significance of the heritability estimates was evaluated through model comparison between models with and without parameter A. For example, if the ACE model was the best-fitting model, the $P$-value was derived by comparing the ACE and CE models; if the best-fitting model was CE, we compared this model with the ACE model to get the $P$-value for the A parameter. We also used the same approach for measuring the statistical significance of the common environmental factor (C). FDR ($q$ = 0.05) was used to correct for multiple comparisons (corrected $P$-values are denoted $P_{FDR}$) in all analyses except for the CCA, where FWER was controlled using a permutation-based procedure (*Winkler et al., 2020*).

## Acknowledgements

AF was supported by the Sylvia and Charles Viertel Foundation, National Health and Medical Research Council (IDs: 1197431 and 1146292), and Australian Research Council (ID: DP200103509). Data were provided in part by OASIS-3: Principal Investigators: T Benzinger, D Marcus, J Morris; NIH P50 AG00561, P30 NS09857781, P01 AG026276, P01 AG003991, R01 AG043434, UL1 TR000448, R01 EB009352. AV-45 doses were provided by Avid Radiopharmaceuticals, a wholly owned subsidiary of Eli Lilly. Data were provided in part by the Human Connectome Project, WU-Minn Consortium (Principal Investigators: David Van Essen and Kamil Ugurbil; 1U54MH091657) funded by the 16 NIH Institutes and Centers that support the NIH Blueprint for Neuroscience Research; and by the McDonnell Center for Systems Neuroscience at Washington University. Data collection and sharing for this project was funded by the Alzheimer's Disease Neuroimaging Initiative (ADNI) (National Institutes of Health Grant U01 AG024904) and DOD ADNI (Department of Defense award number W81XWH-12-2-0012). ADNI is funded by the National Institute on Aging, the National Institute of Biomedical Imaging and

Bioengineering, and through generous contributions from the following: AbbVie, Alzheimer's Association; Alzheimer's Drug Discovery Foundation; Araclon Biotech; BioClinica, Inc; Biogen; Bristol-Myers Squibb Company; CereSpir, Inc; Cogstate; Eisai Inc; Elan Pharmaceuticals, Inc; Eli Lilly and Company; EuroImmun; F Hoffmann-La Roche Ltd and its affiliated company Genentech, Inc; Fujirebio; GE Healthcare; IXICO Ltd.; Janssen Alzheimer Immunotherapy Research & Development, LLC.; Johnson & Johnson Pharmaceutical Research & Development LLC.; Lumosity; Lundbeck; Merck & Co., Inc; Meso Scale Diagnostics, LLC.; NeuroRx Research; Neurotrack Technologies; Novartis Pharmaceuticals Corporation; Pfizer Inc; Piramal Imaging; Servier; Takeda Pharmaceutical Company; and Transition Therapeutics. The Canadian Institutes of Health Research is providing funds to support ADNI clinical sites in Canada. Private sector contributions are facilitated by the Foundation for the National Institutes of Health (https://www.fnih.org/). The grantee organization is the Northern California Institute for Research and Education, and the study is coordinated by the Alzheimer's Therapeutic Research Institute at the University of Southern California. ADNI data are disseminated by the Laboratory for Neuro Imaging at the University of Southern California.

## Additional information

### Competing interests

Alex Fornito: Reviewing editor, *eLife*. Kevin M Aquino: is a scientific advisor and shareholder in BrainKey Inc, a medical image analysis software company. The other authors declare that no competing interests exist.

### Funding

| Funder | Grant reference number | Author |
| --- | --- | --- |
| Sylvia and Charles Viertel Charitable Foundation | Senior Medical Research Fellowship | Alex Fornito |
| National Health and Medical Research Council | 1197431 | Alex Fornito |
| National Health and Medical Research Council | 1146292 | Alex Fornito |
| Australian Research Council | DP200103509 | Alex Fornito |

The funders had no role in study design, data collection and interpretation, or the decision to submit the work for publication.

### Author contributions

Yu-Chi Chen, Conceptualization, Resources, Data curation, Software, Formal analysis, Validation, Investigation, Visualization, Methodology, Writing – original draft, Project administration, Writing – review and editing; Aurina Arnatkevičiūtė, Software, Visualization, Methodology, Writing – review and editing; Eugene McTavish, Software, Methodology; James C Pang, Software, Supervision, Methodology, Writing – review and editing; Sidhant Chopra, Software, Methodology, Visualization; Chao Suo, Resources, Data curation, Software, Methodology; Alex Fornito, Conceptualization, Resources, Supervision, Funding acquisition, Investigation, Visualization, Methodology, Project administration, Writing – review and editing; Kevin M Aquino, Conceptualization, Resources, Software, Supervision, Validation, Investigation, Visualization, Methodology, Project administration, Writing – review and editing

### Author ORCIDs

Yu-Chi Chen http://orcid.org/0000-0001-9167-6417
James C Pang http://orcid.org/0000-0002-2461-2760
Sidhant Chopra http://orcid.org/0000-0003-0866-3477
Alex Fornito http://orcid.org/0000-0001-9134-480X
Kevin M Aquino http://orcid.org/0000-0002-7435-0236

### Ethics

Human subjects: This study only involved subjects from the open-sourced datasets, and all subjects were de-identified by the datasets. Each dataset was approved by its relevant ethics committee and obtained written informed consent from each participant.

### Decision letter and Author response

Decision letter https://doi.org/10.7554/eLife.75056.sa1
Author response https://doi.org/10.7554/eLife.75056.sa2

## Additional files

### Supplementary files

- Supplementary file 1. Wavelength and eigenvalue indices of each eigen-group.
- Transparent reporting form

### Data availability

All data generated or analysed during this study are included in the manuscript. All code and dependent toolboxes used in this study can be found at: https://github.com/cyctbdbw/Shape-Asymmetry-Signature, (copy archived at swh:1:rev:242d90d06e90e7f200182acef46d403f3419a85a). The code of shape-DNA can be found at: http://reuter.mit.edu/software/shapedna/. The OASIS-3 dataset is available under https://www.oasis-brains.org/. The ADNI dataset is available under https://adni.loni.usc.edu. The HCP dataset is available under https://db.humanconnectome.org/.

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
