## [Editor Report]

The article is of interest to scientists who study neuroanatomy or the many behavioral phenotypes that have been proposed be associated with left–right asymmetry of the human brain. The methodology is sophisticated and rigorously applied.

---

## [Decision Letter]

**Decision letter after peer review:**

Thank you for submitting your article "The individuality of shape asymmetries of the human cerebral cortex" for consideration by *eLife*. Your article has been reviewed by 2 peer reviewers, and the evaluation has been overseen by a Reviewing Editor and Christian Büchel as the Senior Editor. The following individual involved in review of your submission has agreed to reveal their identity: Antoine Balzeau (Reviewer #2).

The reviewers have discussed their reviews with one another, and the Reviewing Editor has drafted this to help you prepare a revised submission. The theoretical notions of asymmetry being explored should be more carefully defined and related to past literature, and alternative explanations of some of the results need to be considered. The reviews follow below.

*Reviewer #1 (Recommendations for the authors):*

– It would be very helpful to have improved anatomical representations of the inter-individual differences. For instance, could it be shown what an individual with an extreme eigenvalue at a specific scale would look like, for some of the scales that are argued to be biologically important?

– The results would be strengthened if a measure of bihemispheric (symmetric shape signature) were included, rather than or in addition to unihemispheric shape of the left and right hemisphere separately, to strengthen the conclusions about asymmetry.

– An analysis of a quantitative QC metric could strengthen the paper. For instance, looking at associations of the Euler index from FreeSurfer with SAS across scales may help clarify if there is an association between image quality and these metrics.

– Can the authors clarify that the calculation of SAS is not affected by whether there is a 1:1 correspondence (homology) between left and right hemisphere vertices, which varies between processing pipelines.

*Reviewer #2 (Recommendations for the authors):*

I am pleased to identify myself as Antoine Balzeau. I am a paleoanthropologist, working on the evolution of the human brain. I also work with more "classic" neuroscientists. Please, excuse me if some of my comments or questions appear to be strange for you. I am not able to understand all the mathematic behind your methodology. However, my expertise allows me to comment some of your approaches and ideas, and to ask some questions about the way you deal with bilateral variation. I detail those points below, following the order of the manuscript.

Abstract. Here and later, you mention a lot the context of studies of brain asymmetries. But your study is not really dealing with those aspects. You develop a new tool and find an original application, I agree with this. However, I think that you should improve the way you present the results about bilateral variation and develop implications for morphological and functional aspects. Or only focus on your specific approach and results, that could be enough.

Line 48. You mention DA. However, statement lines 47 and 48 is not completely true. Fluctuating asymmetry is also analysed within a group or a population. DA is only a part of the information obtained within studies of bilateral variation.

Line 51, 52 "regions", "directions", "magnitudes" are not "inconsistent". Different methods give different results about DA and FA in studies of brain bilateral variations. There are consistent results on many specific traits. You cannot compare as a whole the literature and say there is not info from all this amount of research. (I agree with lines 59-61, but it does not justify what you have written above).

Line 62: "little…. Average", this general statement is strange. Again, there are some results showing consistent DA, high absolute asymmetry, antisymmetry….

Line 65-66 FA is not only "the degree to which a given individual departs from the mean". FA is also studied among and between samples, that is the most informative way to discuss developmental instability.

Lines 94-99. I do not understand well this section. I do not see the link with your following sentence. Moreover, I do not think that your study is "a detailed characterization of FA in cerebral shape". At least, I think that you could have said much more on the topic.

Line 101, individual-specific measure of FA. You have to explain this. Why would it be FA? In fact, that is only individual bilateral variation.

Line 106. How can you say that it is done with greater accuracy? You have to demonstrate this.

The end of the introduction is a kind of abstract of what has been done. There are no H0 to be tested, no justification of the approach, no scientific question.

Line 124-125, it is not clear what are the shape descriptors your analyse, nor which aspects of cortical shape asymmetries you study.

Lines 132-145, 150-161, 181-186 should be in the method section.

Line 146-150: methods too, and not clear here why it relates to studies of bilateral variation.

Line 190-191. How do you reach this conclusion? Should be true if DA is present, but how can you compare differences in R-L shapes and shape of an object?

Line 194-197: the conclusion is that one hemisphere does not predict the shape of the other. OK. But why do not you quantify those bilateral variations? Moreover, there is no quantification of absolute asymmetry, DA, or FA. Would not it be possible to extract information about those parameters with you approach? That would be much more informative.

Figures 1 and 2 are more about the methods that true results.

Line 219-223. What is the implication of this? Isn't it related to the fact that the approximation for lower values is too large to extrapolate any information?

Line 226. I do not understand what are identifiability scores, a p = 0, and do not have any idea about the meaning in the variation of the scores between samples. Moreover, why do you do comparison between sample and not for each individual? Finally, why do you have differences between the samples, one with a flat area, differences in shape and values between samples? It seems that you describe global results by sample but never extract something from the descriptions of general results.

Line 230-238: move to the method section.

Line 238-242: what is the implication of this? What does it mean?

243. Why 144 would be a "stable and robust scale"? how can you decide it is 144.

251-253. OK, it works for individual identification. But is it better than any simple comparison of brain shape? Why is it interesting to have tool that allows to identify an individual with 2 mri dataset? Moreover, I would prefer to see more results about bilateral variation thanks to your approach instead of this possibility of identification.

Lines 264-266. You do not demonstrate this supposed higher ability.

Lins 271-272. OK. But what is the implication of this?

Lines 291-295: move to methods. And why did you decide to do this way?

Figure 4: that is the first presentation of bilateral variation. But not test, no description, no quantification of DA, FA, absolute asym…. Why?

Figure 4B, what is the color code? lines 317-324: methods.

Lines 33-335: you mention direction of asymmetries, but there is no test for DA for those parameters.

Figure s5. What are the results for absolute asymmetry?

Lines 344-349: methods.

Line 352-3. Ok, but why is it nteresting to exclude sulcal and gyral features? Isn’t it a too large approximation of the shape? Moreover, you never mention any attempts to test measurement errors, that may have a large impact on FA analyses.

Lines 357-9. Why?

Lines 378-379: what is the evidence that proves this 1/3 relationship?

Discussion

391, in my opinion you have not examined FA.

393: highly personalized, ok, but it is not the most interesting aspect of FA in theory.

Line 394 “most discriminative”, it has not been demonstrated.

Lines 390-398 are an abstract of the paper, not the beginning of a discussion.

Line 401 which “meaningful effects”?

Line 403-404 this image is not appropriate. Line 405-6: ok, but it has to give results that are interpretable, it is not really the case here.

Lines 407 and following: there are several papers on endocranial casts with detailed shape information.

410 which time?

411-412, what is the implication?

416: why are not they “captured reliably”?

419 explain in which aspects it is a “more comprehensive characterization”.

Lines 434-440: No link with your study and results.

Line 463: ok, but please elaborate a bit.

Lines 474-475: this is a rather naïve statement. Please explain or delete.

Conclusion: to be removed, it is an abstract.

Methods

503: how do you determine those errors?

On the reverse, you do not quantify your own methodological error. Have you tried to repeat your methods? What is the effect of imaging resolution (always important when working on bilateral variation)?

Line 759 what are the “morphological measurements” that you mention here?

Line 770: what are the asym-index measurements?

Line 798: what is reduced?

---

## [Author Response]

Reviewer #1 (Recommendations for the authors):1. It would be very helpful to have improved anatomical representations of the inter-individual differences. For instance, could it be shown what an individual with an extreme eigenvalue at a specific scale would look like, for some of the scales that are argued to be biologically important?

Thank you for this valuable suggestion. We have included an additional figure (Figure 1—figure supplement 1) that shows cortical surface reconstructions at different scales for varying values of the SAS.

2. The results would be strengthened if a measure of bihemispheric (symmetric shape signature) were included, rather than or in addition to unihemispheric shape of the left and right hemisphere separately, to strengthen the conclusions about asymmetry.

Thank you for this comment. In Figure 2 (A) to (C), we showed that the subject identifiability of the SAS is not only higher than unihemispheric shape (either left or right hemisphere) but also higher than combining the shapes of both hemispheres (concatenating the left and right hemispheres). For this reason, the subsequent analyses focus on the SAS. We also note that the SAS can readily accommodate cases of pure symmetry, and in such cases, the SAS would be equal to zero. We now include a demonstration of the point in the new Figure 2—figure supplement 2.

3. An analysis of a quantitative QC metric could strengthen the paper. For instance, looking at associations of the Euler index from FreeSurfer with SAS across scales may help clarify if there is an association between image quality and these metrics.

Thank you for raising this important issue. We have checked that the Euler number of the cortical surfaces, which is recommended as a quantitative index of image quality (1), is unrelated to the SAS, indicating that image quality has a limited influence on our measure of shape asymmetry.

4. Can the authors clarify that the calculation of SAS is not affected by whether there is a 1:1 correspondence (homology) between left and right hemisphere vertices, which varies between processing pipelines.

Thank you for raising this important point. Before the calculation of SAS, we calculated the eigenvalues of the Laplace-Beltrami Operator (LBO) from the left and right hemispheres independently. The LBO is not directly computed on the vertices of surface mesh but on the underlying manifold of the surface (14). In other words, the LBO quantifies the shape of the entire hemisphere, and thus, the SAS is not affected by the 1:1 correspondence between the left and right hemisphere vertices.

Reviewer #2 (Recommendations for the authors):I am pleased to identify myself as Antoine Balzeau. I am a paleoanthropologist, working on the evolution of the human brain. I also work with more “classic” neuroscientists. Please, excuse me if some of my comments or questions appear to be strange for you. I am not able to understand all the mathematic behind your methodology. However, my expertise allows me to comment some of your approaches and ideas, and to ask some questions about the way you deal with bilateral variation. I detail those points below, following the order of the manuscript.

Thank you for your detailed feedback and suggestions.

1. Abstract. Here and later, you mention a lot the context of studies of brain asymmetries. But your study is not really dealing with those aspects. You develop a new tool and find an original application, I agree with this. However, I think that you should improve the way you present the results about bilateral variation and develop implications for morphological and functional aspects. Or only focus on your specific approach and results, that could be enough.

We interpret this comment to be focused on the Abstract. We note that studies on asymmetry provide an important context and motivation for our approach; namely, to develop a method that can disentangle contributions from size and shape across multiple spatial resolution scales. To better develop the background in the Abstract, we have now amended the text so that it reads:

“Asymmetries of the cerebral cortex are found across diverse phyla and are particularly pronounced in humans, with important implications for brain function and disease. However, many prior studies have confounded asymmetries due to size with those due to shape. Here, we introduce a novel approach to characterize asymmetries of the whole cortical shape, independent of size, across different spatial frequencies using magnetic resonance imaging data in three independent datasets…”.

We have also made several further amendments to other sections of the text to clarify our reporting on prior results. These are detailed in our responses to subsequent comments.

2. Line 48. You mention DA. However, statement lines 47 and 48 is not completely true. Fluctuating asymmetry is also analysed within a group or a population. DA is only a part of the information obtained within studies of bilateral variation.

To the best of our knowledge, fluctuating asymmetry (FA) and directional asymmetry (DA) are not consistently defined across studies and across disciplines. In our original manuscript, we followed the definitions of FA and DA in references (15-17). However, we agree that using these terms may confuse some readers with different backgrounds. We have therefore refrained from using these terms in our revised manuscript and instead used population-level/population-based asymmetry and individual-specific asymmetry. These two terms more clearly articulate the type of asymmetry being considered.

3. Line 51, 52 “regions”, “directions”, “magnitudes” are not “inconsistent”. Different methods give different results about DA and FA in studies of brain bilateral variations. There are consistent results on many specific traits. You cannot compare as a whole the literature and say there is not info from all this amount of research. (I agree with lines 59-61, but it does not justify what you have written above).Line 62: “little…. Average”, this general statement is strange. Again, there are some results showing consistent DA, high absolute asymmetry, antisymmetry….

We agree. We now cite examples of consistent results and have revised Lines 57 to 66 of the revised manuscript:

“Asymmetries in brain organization are often considered at an average level across a population of individuals (7, 18-21). These population-based asymmetry features have been studied extensively and are thought to have important implications for both functional lateralization and abnormal brain function in a wide range of psychiatric and neurological diseases (20, 22-25). For example, the planum temporale of the left hemisphere, which encompasses Wernicke’s area, has been consistently shown to be larger than the right for most healthy individuals (21, 26-28), and patients with schizophrenia often show reduced leftward asymmetry in planum temporale compared to healthy individuals (29-31). However, many findings with respect to asymmetries of specific brain regions have been inconsistent in terms of the directions and magnitudes of asymmetry observed (18, 19, 23).”

We have also revised Lines 74 to 78 of the revised manuscript:

“Despite some consistent asymmetry features across the population (21, 26-28), there is also considerable individual variability around population means, with many people often showing little or even reversed asymmetries relative to the prevalent pattern of the population (sometimes also referred to as anti-symmetry) (16, 32-34).”

4. Line 65-66 FA is not only “the degree to which a given individual departs from the mean”. FA is also studied among and between samples, that is the most informative way to discuss developmental instability.

As per our response to *Comment 2*, we deleted all references to FA and DA, and amended the following sentences in Lines 80 to 85 of the revised manuscript:

“Populational-level asymmetries are hypothesized to have a genetic basis (16, 18, 33, 35-40), whereas individual-specific asymmetries, which describe the way in which a given individual departs from the population mean, may reflect environmental influences, developmental plasticity, or individual-specific genetic perturbations (16, 17, 33, 35-38). Notably, cortical asymmetries of the human brain are more variable across individuals than other primates, at both regional and global hemispheric levels (16, 32).”

5 Lines 94-99. I do not understand well this section. I do not see the link with your following sentence. Moreover, I do not think that your study is “a detailed characterization of FA in cerebral shape”. At least, I think that you could have said much more on the topic.

The introduction outlines the general principles of shape-DNA and our spectral approach, which provides a context for our aims and for the results that follow. The *Materials and methods* section outline in detail how the approach is implemented. We now include in Lines 108 to 115 of the revised manuscript an indication to readers that further details are provided in the *Materials and methods* section. We have also amended the statement about FA so that it now reads:

“The spectral analysis provides a comprehensive description of the intrinsic geometry of a given object, akin to a “Shape-DNA” (see *Materials and methods*). (14). The application of such Shape-DNA analysis to human magnetic resonance imaging (MRI) data has shown that shape properties of cortical and subcortical structures have superior sensitivity compared to traditional, size-based measures for identifying individual subjects (41), for classifying and predicting the progress of psychiatric and neurological diseases (42, 43), and for detecting genetic influences on brain structure (44, 45). However, a detailed characterization of individual-specific asymmetries in cerebral shape is lacking.”

6. Line 101, individual-specific measure of FA. You have to explain this. Why would it be FA? In fact, that is only individual bilateral variation.

As per our responses to *Comments 1 and 2,* we no longer use the term FA and instead use individual-specific asymmetry, which we define in Lines 655 to 673 of the revised manuscript. Moreover, as per our response to Reviewer 1’s *Comment 2*, we have provided an additional figure (Figure 1—figure supplement 1) to show that the SAS value of a perfectly symmetric brain is zero, and it increases for more asymmetric brains.

7. Line 106. How can you say that it is done with greater accuracy? You have to demonstrate this.

This is one of our key findings. In Figure 3, we show that the SAS is more accurate at identifying individuals than shape descriptors of individual hemispheres, asymmetries in traditional size-based descriptors, or patterns of inter-regional functional connectivity.

8. The end of the introduction is a kind of abstract of what has been done. There are no H0 to be tested, no justification of the approach, no scientific question.

We apologize for the lack of clarity. Our motivation follows from the idea that the eigen decomposition of the LBO can be used as a form of shape-DNA (14); i.e., it can be used to identify individual brains (41). As individual-specific asymmetries appear to be highly variable in humans (16, 32), our primary hypothesis was that the SAS would be associated with greater identifiability than measures of unihemispheric cerebral shape, volume, or function. The null hypothesis then is that the SAS does not show greater identifiability than unihemispheric measures. We secondarily explored associations with gender, handedness and cognition, given the extant literature linking various forms of cerebral asymmetry to these factors. Finally, we tested whether the SAS is more strongly influenced by environmental factors, given arguments that individual-specific asymmetries may be primarily driven by environmental or developmental plasticity (16, 17, 33, 35-38). We have amended Lines 116 to 130 of the revised manuscript to more clearly outline our aims and hypotheses. We underline our hypotheses here for convenience:

“Here, we introduce methods for constructing an individual-specific measure of cortical asymmetry, called the shape asymmetry signature (SAS; see *Materials and methods*). The SAS characterizes pure shape asymmetries of the whole cortical surface, independent of variations in size, across a spectrum of spatial scales. We apply this methodology to three independent longitudinal datasets to test the hypothesis that cortical shape asymmetry is a highly personalized and robust feature that can identify individuals, akin to a cortical asymmetry fingerprint. We then use the identifiability values to identify optimal spatial scales at which robust individual differences are most salient. We also compare the identifiability of the SAS and shape descriptors of individual hemispheres, asymmetries in traditional size-based descriptors, or patterns of inter-regional functional connectivity (so-called connectome fingerprinting (46)) to test the hypothesis that the SAS is a more individually unique property of brain organization than unihemispheric and functional properties. We further elucidate the relationships between the SAS and sex, handedness, as well as cognitive performance across multiple tasks. Finally, we test the hypothesis that individual-specific asymmetry features are largely driven by environmental influences using classical heritability modelling of twin data.”

9. Line 124-125, it is not clear what are the shape descriptors your analyse, nor which aspects of cortical shape asymmetries you study.

Lines 124 to 125 are just the first sentence of the first paragraph of the *Results section* and are intended to orient readers. The descriptions come below. We have now amended the Lines 133 to 135 of the revised manuscript so that it reads:

“To understand how cortical shape asymmetries vary across individuals, we examined the degree to which different cortical shape descriptors (defined below) can be used to identify individual brains from a large sample of T1-weighted magnetic resonance images (MRIs).”

10. Lines 132-145, 150-161, 181-186 should be in the method section.

We agree that these lines contain methodological detail. However, given the *eLife* format, which places Methods at the end of the paper, we contend that it is essential to include this level of methodological detail in the *Results section* to provide sufficient context for interpreting the *Results.*

11. Line 146-150: methods too, and not clear here why it relates to studies of bilateral variation.

We performed the identifiability analysis to investigate whether the cortical shape asymmetry is an individualized feature of human brain. This is an excellent and widely-used (47, 48) method for determining just how individually specific such asymmetries are. As per our response to *Comment 10*, we include this level of methodological detail here to provide context for interpreting the *Results section*.

12. Line 190-191. How do you reach this conclusion? Should be true if DA is present, but how can you compare differences in R-L shapes and shape of an object?

Identifiability quantifies person-specific variability. If a phenotype has high identifiability, then it is highly specific to individuals. Given that the SAS is associated with higher identifiability than shape descriptors of unilateral hemispheres and descriptors obtained from combining both left and right hemispheres, we can conclude that the SAS captures greater individual variability. The eigen decomposition of the LBO allows us to characterize and compare shapes, as detailed in the *Materials and methods* section.

13. Line 194-197: the conclusion is that one hemisphere does not predict the shape of the other. OK. But why do not you quantify those bilateral variations? Moreover, there is no quantification of absolute asymmetry, DA, or FA. Would not it be possible to extract information about those parameters with you approach? That would be much more informative.

The SAS is precisely designed to quantify hemispheric differences in shape. It is possible To compute an absolute version of the SAS, but this removes specific individually unique information. We confirm this in the analysis in Author response image 1, which shows that the absolute value of SAS is associated with lower identifiability than our original SAS.

**Author response image 1. sa2fig1:** 

14. Figures 1 and 2 are more about the methods that true results.

Figure 1 provides a visual schematic of our methodological approach and is essential for contextualizing the *Results section*. It is common to include such figures in our field. Figure 2 presents some of our key findings, comparing the identifiability of the SAS with other shape measures and truncating the spatial scales to represent the maximal identifiability.

15. Line 219-223. What is the implication of this? Isn’t it related to the fact that the approximation for lower values is too large to extrapolate any information?

The implications of the maximal subject identifiability at coarse scales are discussed in the *Identifiability of cortical shape asymmetry is maximal at coarse scales* section of the *Discussion*. We repeatedly evaluated the identifiability using up to 1000 eigenvalues. There is no reason to think that this value is too large to extrapolate useful information. For example, Germanaud et al. (49) used 5000 eigenfunctions (corresponding to 5000 eigenvalues) to examine cortical curvature. As per our response to *Comment 1* of Reviewers 1’s *Public Review*, we have now included details about the potential role of measurement error at finer spatial scales in Lines 266 to 269, 437 to 439, and Figure 2—figure supplement 2.

16. Line 226. I do not understand what are identifiability scores, a p = 0, and do not have any idea about the meaning in the variation of the scores between samples. Moreover, why do you do comparison between sample and not for each individual? Finally, why do you have differences between the samples, one with a flat area, differences in shape and values between samples? It seems that you describe global results by sample but never extract something from the descriptions of general results.

Thank you for your questions. We have amended this sentence in Lines 240 to 242 of the revised manuscript:

“At these scales, the subject identifiability scores were 4.93 (*P* = 0; estimated by permutation; see *Statistical analysis* section in *Materials and methods* for details) for OASIS-3 and 5.03 (*P* = 0) for ADNI.”

To compute the identifiability score, we estimate correlations between all possible pairs of time 1 and time 2 phenotypes (e.g., SAS scores). The identifiability score is essentially the difference between the correlations observed for the same person at both time points and different people at the two time points. In other words, identifiability is high if a person *i*’s time 1 scan is more correlated with his/her scan at time 2 than with someone else’s scan. The formal definition of identifiability is provided in the *Subject Identification* section of the *Materials and methods.*

We perform statistical inference on the identifiability scores by permuting the participant identities. This gives us a distribution of identifiability scores under the null hypothesis. The *p*-value is then given by the fraction of null values that exceed the observed identifiability score. A p = 0 means that no null values exceeded the observed score. This approach is necessary as no parametric statistic is available for inference on identifiability scores, which is commonly used in the literature (e.g., (46, 48)). This procedure is outlined in detail in the *Materials and methods* section.

We examine three datasets to test the robustness of our findings; i.e., we replicate our results in independent samples. It does not really make sense to combine the datasets since they represent different cohorts measured with different scanners. The fact that we observe such consistent findings across all cohorts speaks to the robustness of our results. We focus on the general, consistent trends across the three cohorts in the *Discussion section*.

17. Line 230-238: move to the method section.

This description is essential for readers to understand the full context of the results that follow, since the *Materials and methods* are presented at the end of the paper.

18. Line 238-242: what is the implication of this? What does it mean?

The eigen-groups provide a way to group eigenvalues into spatial variations of similar wavelengths, in direct analogy to spherical harmonics, as detailed in Lines 730 to 750 of the revised manuscript. We also explain the implication and meaning of eigen-groups in Lines 259 to 266 of the revised manuscript:

“Thus, the first 144 eigenvalues represent a stable and robust characteristic scale at which individual uniqueness in cortical shape asymmetry is strongest. The 11^th^ group corresponds to a wavelength of approximately 37 mm in the case of the population-based template (fsaverage in FreeSurfer; Appendix 1—Table 1 shows the corresponding wavelengths of the first 14 eigen–groups; Figure 2G shows the spatial scales corresponding to the cumulative eigen-groups). A reconstruction of the cortical surface using the first 144 eigenfunctions is shown in Figure 2H.

The reconstruction captures shape variations at a coarse scale, representing major primary and secondary sulci, but with minimal additional details.”

19. 243. Why 144 would be a "stable and robust scale"? how can you decide it is 144.251-253. OK, it works for individual identification. But is it better than any simple comparison of brain shape? Why is it interesting to have tool that allows to identify an individual with 2 mri dataset? Moreover, I would prefer to see more results about bilateral variation thanks to your approach instead of this possibility of identification.

We choose 144 because the first 144 eigenvalues correspond to the first 11 eigen-groups, and across the three datasets peak or near-peak identifiability is observed within the 11^th^ eigen-group. It is stable and robust because we obtain the same result across three independent datasets. We show in Figures 2 and 3 that the SAS offers stronger identifiability than unihemispheric shape descriptors and asymmetries of cortical volume, cortical thickness, or surface area, so it is better than simpler shape and size-related descriptors. We focus on identifiability as a way of quantifying how individually unique shape asymmetries are. This question speaks to how variable such asymmetries are across the population. Note that the eigenvalues of the LBO provide a compact representation of shape variation at different scales, and the corresponding eigenfunctions (examples of which are shown in Figures 2 (G) and (H)) represent the basis set of spatial patterns. The SAS thus offers an efficient summary of cortical asymmetries. In the revised manuscript, we now include Figure 1—figure supplement 1, which more clearly illustrates the correspondence between different SAS values and cortical shape at different spatial scales. After determining the spatial scales with the optimal subject identifiability, we linked the SAS to sex and cognitive function, as discussed in Lines 309 to 361 of the revised manuscript.

20. Lines 264-266. You do not demonstrate this supposed higher ability.

We have demonstrated that “individual variability in brain anatomy is higher when considering asymmetries in cortical shape compared to more traditional size-based morphological descriptors” in Figure 3 (A) to (B) and Figure 3—figure supplement 1. Specifically, Figure 3 (A) shows that identifiability scores for the SAS are higher than those obtained for asymmetries based on cortical surface area (identifiability score = 0.81), volume (identifiability score = 0.66), and thickness (identifiability score = 0.33). Figure 3—figure supplement 1 demonstrates that the identifiability scores calculated from the surface area normalized SAS are generally higher than the scores calculated using native eigenvalues and eigenvalues with volume normalization (but without surface area normalization) for individual hemispheres, the combination of both hemispheres, and asymmetry.

21. Lines 271-272. OK. But what is the implication of this?

Functional connectivity has been shown to be a highly personalized brain phenotype, akin to a neuro-fingerprint, in numerous studies (46, 48, 50, 51). Here, we show that the SAS has an even better subject identifiability than this widely studied feature. It indicates that cortical shape asymmetries are highly specific to individual brains.

22. Lines 291-295: move to methods. And why did you decide to do this way?

As detailed above, *eLife* presents the *Materials and methods* section at the end of the article, hence this detail is required to provide sufficient context for the results that follow. Using a GLM with permutation testing in this context is standard in the field for this sort of analysis. We use permutation testing to infer statistical significance because the subjects in the HCP dataset are not independent; some of them are from the same twin pairs or non-twin siblings.

23. Figure 4: that is the first presentation of bilateral variation. But not test, no description, no quantification of DA, FA, absolute asym…. Why?

The SAS quantifies cortical shape asymmetries. Its effects are quantified extensively in Figures 2 to 4. The brain pictures in Figure 4 (B) simply show the eigenfunctions that correspond to the eigenvalues at which significant associations were identified. It is intended simply to show the gradients of spatial variation associated with the implicated eigenvalues. Detailed descriptions of eigenvalues and eigenfunctions are provided in the *Spectral shape analysis* section of *Materials and methods.* To clarify this further, we have amended Lines 332 to 334:

“(B) The corresponding eigenfunction of each eigenvalue in panel (A) that shows the gradients of spatial variation on a population-based template.”

24. Figure 4B, what is the color code?

It is the value of each cortical point for each eigenfunction. We have now added colorbars in Figure 4 (B):

25. Lines 317-324: methods.

As detailed above, it is difficult for readers to understand this section without briefly describing the methods first.

26. Lines 33-335: you mention direction of asymmetries, but there is no test for DA for those parameters.

Thank you for your question. To avoid confusion, we have changed the terms in the revised manuscript, replacing DA with population-based asymmetry to refer to the asymmetry map across a population. The direction of the asymmetry is captured by the sign of the SAS, such that negative values indicate rightward asymmetry and positive values indicate leftward asymmetry. The direction of asymmetry is specific to individuals. Since the canonical correlation analysis measures individual variations in both the shape asymmetry and cognitive function, our interpretation is correct.

27. Figure s5. What are the results for absolute asymmetry?

We find that when using absolute asymmetry (i.e., the eigen-groups from the SAS with absolute value), the canonical correlation analysis (CCA) results are not significant (*P_FWER_* > 0.6). As mentioned in our response to *Comment 13*, absolute asymmetry also shows inferior identifiability to our SAS measure.

28. Lines 344-349: methods.

As per above, this description is necessary for context.

29. Line 352-3. Ok, but why is it interesting to exclude sulcal and gyral features? Isn't it a too large approximation of the shape? Moreover, you never mention any attempts to test measurement errors, that may have a large impact on FA analyses.

To clarify, we do not explicitly exclude sulcal and gyral features. Our findings tell us that identifiability is optimal at spatial scales that are coarser than fine-grained anatomical features such as some specific sulci and gyri. This is interesting because it indicates that highly unique and individual properties of brain shape asymmetry are found at coarse scales. As indicated in our response to *Comments 1* and *3* of the *Public Review* of Reviewer 1, we revised the manuscript to now include a characterization of measurement error across scales and show that the SAS is robust to variations in scan quality, but that fine spatial scales may be too noisy to inform identifiability.

30. Lines 357-9. Why?

As shown in Figure 6, strong heritability is only found within the first six eigen-groups, corresponding to wavelengths greater than or equal to ~65 mm. The insets show the corresponding spatial scales of the reconstructions.

31. Lines 378-379: what is the evidence that proves this 1/3 relationship?

Figure 6 (C) shows that the of the SAS never exceeds 0.30.

32. Discussion391, in my opinion you have not examined FA.393: highly personalized, ok, but it is not the most interesting aspect of FA in theory.

We have revised Lines 410 to 414 to remove all the FA terms in the revised manuscript:

“Here, we employed a multiscale approach to quantify individualized shape asymmetries of the human cerebral cortex. We found that cortical shape asymmetries were highly personalized and robust, with shape asymmetries at coarse spatial scales being the most discriminative among individuals, showing differences between males and females, and correlating with individual differences in cognition.”

33. Line 394 "most discriminative", it has not been demonstrated.

We have demonstrated this in Figure 2 (**A**) to (**C**) and Lines 201 to 211 of the revised manuscript by showing that the SAS has the highest identifiability.

34. Lines 390-398 are an abstract of the paper, not the beginning of a discussion.

In our field it is common to begin the *Discussion section* with a brief summary of findings.

35. Line 401 which "meaningful effects"?

We have revised this sentence in Lines 420 to 421 of the revised manuscript so that it now reads:

“These approaches may ignore meaningful effects (i.e., properties that are individually unique and correlated with cognition) at coarser spatial scales.”

36. Line 403-404 this image is not appropriate. Line 405-6: ok, but it has to give results that are interpretable, it is not really the case here.

Lines 403 to 404 of the original manuscript state that:

“Our approach is akin to studying seismic waves of earthquakes with different wave frequencies at the global tectonic scale, instead of focusing on a particular city.”

This sentence provides an intuitive understanding for our multiscale analysis compared to traditional point-wise analyses. In fact, our approach draws on mathematical tools established for over a century and used in diverse fields of physics and engineering to characterize spatial properties of different systems. We contend that our results are indeed interpretable, as we outlined in the manuscript, but the interpretation is just different to that offered by traditional neuroscientific analyses. As per our response to Reviewer 1’s *Comment 2,* the revised manuscript now includes Figure 2—figure supplement 2 to aid interpretation.

37. Lines 407 and following: there are several papers on endocranial casts with detailed shape information.

We agree that there are several papers on endocranial casts, but most of those papers simply quantified the linear dimensions of Broca’s cap or petalia and did not characterize the actual 3D shape of the entire endocranial casts or brain surface (16). Many of those papers also ignored individual variations in actual shapes, focusing only on average patterns across a population (16). We have revised the sentence in Line 427 of the revised manuscript:

“Few studies have assessed individual variations in shape at coarse scales.”

38. 410 which time?

Kong et al. (6) used the HCP test-retest dataset, which is one of the datasets we used in this study, and the intervals between time 1 and time 2 were about one to 11 months.

39. 411-412, what is the implication?

We have added an additional sentence in Lines 432 to 434 of the revised manuscript:

“Wachinger et al. (21) used shape descriptors at coarse scales derived from the eigenvalues of the LBO for all brain structures to achieve accurate subject identification. Taken together with our findings, these results indicate that coarse features of cortical shape are highly personalized and unique to individuals.”

40. 416: why are not they "captured reliably"?

Line 416 states that:

“It is possible that these fine-grained features are not captured reliably, meaning that they contribute noise to identifiability analyses.”

In brief, we provide an additional figure (Figure 2—figure supplement 2), which used one subject’s images from two MRI sessions. The figure shows that more inter-sessional differences (quantified by the Euclidean distances between the vertices) between time 1 and time 2 images are more observable at finer spatial scales. In other words, information at finer scales was more vulnerable to measurement noise or subtle fluctuations in volume affecting shape.

41. 419 explain in which aspects it is a "more comprehensive characterization".

We have revised the sentence in Lines 441 to 442 of the revised manuscript:

“Our multiscale approach affords a more comprehensive characterization of shape variations across multiple spatial scales.”

42. Lines 434-440: No link with your study and results.

We found that cortical shape asymmetry is mainly driven by unique environmental effects. This section outlines potential ways in which this may occur.

43. Line 463: ok, but please elaborate a bit.

Thank you for your suggestions, but our scale-specific asymmetry analysis is a new approach with no existing references to cite and the specific mechanism is beyond the scope of the present study. For this reason, we say that it requires further investigation.

44. Lines 474-475: this is a rather naïve statement. Please explain or delete.

This sentence briefly summarizes the preceding paragraph and is merely based on our results. Specifically, at the beginning of this paragraph (Lines 490 to 493 of the revised manuscript), we mentioned:

“We found that a greater leftward asymmetry across a wide range of spatial scales, corresponding to wavelengths of approximately 37, 75, 95, and 170 mm, and greater rightward asymmetry at wavelengths of approximately 40, 55, 65, and 120 mm, are associated with better performance across nearly all cognitive measures considered.

Based on our result, we concluded that different scales may have different relationships with cognition. For this reason, a multiscale approach can reveal associations between brain and behavior that may not be detected with single-scale approaches.”

45. Conclusion: to be removed, it is an abstract.

We believe that it is useful to end the paper with a brief summary of findings. We have renamed this section as *Summary* instead of Conclusions.

46. Methods503: how do you determine those errors?On the reverse, you do not quantify your own methodological error. Have you tried to repeat your methods? What is the effect of imaging resolution (always important when working on bilateral variation)?

Line 503 reads:

“We excluded six subjects whose SAS was an outlier in at least one of those sessions due to poor image quality and major errors in image segmentation. These subjects had more than two eigenvalues of the first 200 eigenvalues that departed from the population mean values by more than four standard deviations.”

Segmentation errors are determined via manual inspection of all FreeSurfer outputs. FreeSurfer is a widely validated technique (52-54), and errors are typically determined by visual inspection. Moreover, the eigen decomposition of the LBO is robust to variations in mesh resolution and image quality (4). As outlined in our responses to Reviewer 1’s *Recommendations for the authors* and *Public Review Comment 3*, we now also show that the Euler number from FreeSurfer (1), a quantitative metric of image quality, was not associated with the SAS. We have repeated the SAS on three independent datasets to show that it quantified individualized brain features.

47. Line 759 what are the "morphological measurements" that you mention here?

We now explain "morphological measurements" in Lines 789 to 790 of the revised manuscript:

“In this study, we adjusted the asymmetry index for the mean of each morphological measurement, such as the asymmetry of cortical thickness, volume, and surface area.”

48. Line 770: what are the asym-index measurements?

The asymmetry-index measurements correspond to the asymmetry of cortical thickness, surface area, and volume, all of which were quantified via the asymmetry index. We have revised this sentence in Lines 801 to 803 of the revised manuscript:

“We created one vector per size-related measure that quantified the asymmetry index per subject and then used these asymmetry indices in the subject identifiability analyses.”

49. Line 798: what is reduced?

As indicated in Lines 837 to 841 of the revised manuscript, we used PCA to reduce the dimensionality of the cognitive data, replacing 13 different cognitive test scores with 4 principal components explaining 80% of the test variance. An advantage of this approach is that the components are orthogonal, which mitigates biases in the CCA. We reduced the dimensionality of the SAS values from 143 eigenvalues to 11 by taking the mean of the first 11 eigen-groups. Our rationale for using eigen-groups is outlined in detail in Lines 730 to 750 of the revised manuscript.

References

1. Rosen AFG, Roalf DR, Ruparel K, Blake J, Seelaus K, Villa LP, et al. Quantitative assessment of structural image quality. Neuroimage. 2018;169:407-18.

2. Morgan SE, Seidlitz J, Whitaker KJ, Romero-Garcia R, Clifton NE, Scarpazza C, et al. Cortical patterning of abnormal morphometric similarity in psychosis is associated with brain expression of schizophrenia-related genes. Proc Natl Acad Sci U S A. 2019;116(19):9604-9.

3. Kaufmann T, van der Meer D, Doan NT, Schwarz E, Lund MJ, Agartz I, et al. Common brain disorders are associated with heritable patterns of apparent aging of the brain. Nat Neurosci. 2019;22(10):1617-23.

4. Reuter M. Hierarchical Shape Segmentation and Registration via Topological Features of Laplace-Beltrami Eigenfunctions. International Journal of Computer Vision. 2009;89(2-3):287-308.

5. Maingault S, Tzourio-Mazoyer N, Mazoyer B, Crivello F. Regional correlations between cortical thickness and surface area asymmetries: A surface-based morphometry study of 250 adults. Neuropsychologia. 2016;93(Pt B):350-64.

6. Kong XZ, Postema M, Schijven D, Castillo AC, Pepe A, Crivello F, et al. Large-Scale Phenomic and Genomic Analysis of Brain Asymmetrical Skew. Cereb Cortex. 2021.

7. Deep-Soboslay A, Hyde TM, Callicott JP, Lener MS, Verchinski BA, Apud JA, et al. Handedness, heritability, neurocognition and brain asymmetry in schizophrenia. Brain. 2010;133(10):3113-22.

8. Dragovic M. Categorization and validation of handedness using latent class analysis. Acta Neuropsychiatr. 2004;16(4):212-8.

9. Narr KL, Bilder RM, Luders E, Thompson PM, Woods RP, Robinson D, et al. Asymmetries of cortical shape: Effects of handedness, sex and schizophrenia. Neuroimage. 2007;34(3):939-48.

10. Vlachos F, Avramidis E, Dedousis G, Katsigianni E, Ntalla I, Giannakopoulou M, et al. Incidence and Gender Differences for Handedness among Greek Adolescents and Its Association with Familial History and Brain Injury Research in Psychology and Behavioral Sciences. 2013;1(1):6-10.

11. Perlaki G, Horvath R, Orsi G, Aradi M, Auer T, Varga E, et al. White-matter microstructure and language lateralization in left-handers: a whole-brain MRI analysis. Brain Cogn. 2013;82(3):319-28.

12. Oldfield RC. The assessment and analysis of handedness: the Edinburgh inventory. Neuropsychologia. 1971;9(1):97-113.

13. Willems RM, Van der Haegen L, Fisher SE, Francks C. On the other hand: including left-handers in cognitive neuroscience and neurogenetics. Nat Rev Neurosci. 2014;15(3):193-201.

14. Reuter M, Wolter FE, Peinecke N. Laplace–Beltrami spectra as ‘Shape-DNA’ of surfaces and solids. Computer-Aided Design. 2006;38(4):342-66.

15. Yeo RA, Ryman SG, Pommy J, Thoma RJ, Jung RE. General cognitive ability and fluctuating asymmetry of brain surface area. Intelligence. 2016;56:93-8.

16. Neubauer S, Gunz P, Scott NA, Hublin JJ, Mitteroecker P. Evolution of brain lateralization: A shared hominid pattern of endocranial asymmetry is much more variable in humans than in great apes. Sci Adv. 2020;6(7):eaax9935.

17. Nadig A, Seidlitz J, McDermott CL, Liu S, Bethlehem R, Moore TM, et al. Morphological integration of the human brain across adolescence and adulthood. Proc Natl Acad Sci U S A. 2021;118(14).

18. Kong XZ, Mathias SR, Guadalupe T, Group ELW, Glahn DC, Franke B, et al. Mapping cortical brain asymmetry in 17,141 healthy individuals worldwide via the ENIGMA Consortium. Proc Natl Acad Sci U S A. 2018;115(22):E5154-E63.

19. Kong XZ, Postema MC, Guadalupe T, de Kovel C, Boedhoe PSW, Hoogman M, et al. Mapping brain asymmetry in health and disease through the ENIGMA consortium. Hum Brain Mapp. 2020.

20. Postema MC, van Rooij D, Anagnostou E, Arango C, Auzias G, Behrmann M, et al. Altered structural brain asymmetry in autism spectrum disorder in a study of 54 datasets. Nat Commun. 2019;10(1):4958.

21. Toga AW, Thompson PM. Mapping brain asymmetry. Nat Rev Neurosci. 2003;4(1):37-48.

22. Esteves M, Ganz E, Sousa N, Leite-Almeida H. Asymmetrical Brain Plasticity: Physiology and Pathology. Neuroscience. 2020.

23. Plessen KJ, Hugdahl K, Bansal R, Hao X, Peterson BS. Sex, age, and cognitive correlates of asymmetries in thickness of the cortical mantle across the life span. J Neurosci. 2014;34(18):6294-302.

24. Cai Y, Liu J, Zhang L, Liao M, Zhang Y, Wang L, et al. Grey matter volume abnormalities in patients with bipolar I depressive disorder and unipolar depressive disorder: a voxel-based morphometry study. Neurosci Bull. 2015;31(1):4-12.

25. Fling BW, Dutta GG, Schlueter H, Cameron MH, Horak FB. Associations between Proprioceptive Neural Pathway Structural Connectivity and Balance in People with Multiple Sclerosis. Front Hum Neurosci. 2014;8:814.

26. Gunturkun O, Strockens F, Ocklenburg S. Brain Lateralization: A Comparative Perspective. Physiol Rev. 2020;100(3):1019-63.

27. Royer C, Delcroix N, Leroux E, Alary M, Razafimandimby A, Brazo P, et al. Functional and structural brain asymmetries in patients with schizophrenia and bipolar disorders. Schizophr Res. 2015;161(2-3):210-4.

28. Takao H, Abe O, Yamasue H, Aoki S, Sasaki H, Kasai K, et al. Gray and white matter asymmetries in healthy individuals aged 21-29 years: a voxel-based morphometry and diffusion tensor imaging study. Hum Brain Mapp. 2011;32(10):1762-73.

29. Clark GM, Crow TJ, Barrick TR, Collinson SL, James AC, Roberts N, et al. Asymmetry loss is local rather than global in adolescent onset schizophrenia. Schizophr Res. 2010;120(1-3):84-6.

30. Ratnanather JT, Poynton CB, Pisano DV, Crocker B, Postell E, Cebron S, et al. Morphometry of superior temporal gyrus and planum temporale in schizophrenia and psychotic bipolar disorder. Schizophr Res. 2013;150(2-3):476-83.

31. Corballis MC. Early signs of brain asymmetry. Trends Cogn Sci. 2013;17(11):554-5.

32. Gomez-Robles A, Hopkins WD, Sherwood CC. Increased morphological asymmetry, evolvability and plasticity in human brain evolution. Proc Biol Sci. 2013;280(1761): 115-26.

33. Gomez-Robles A, Hopkins WD, Schapiro SJ, Sherwood CC. The heritability of chimpanzee and human brain asymmetry. Proc Biol Sci. 2016;283(1845).

34. Corballis MC, Haberling IS. The Many Sides of Hemispheric Asymmetry: A Selective Review and Outlook. J Int Neuropsychol Soc. 2017;23(9-10):710-8.

35. Sherwood CC, Gómez-Robles A. Brain Plasticity and Human Evolution. Annual Review of Anthropology. 2017;46(1):399-419.

36. Francks C. Exploring human brain lateralization with molecular genetics and genomics. Ann N Y Acad Sci. 2015;1359:1-13.

37. de Kovel CGF, Lisgo SN, Fisher SE, Francks C. Subtle left-right asymmetry of gene expression profiles in embryonic and foetal human brains. Sci Rep. 2018;8(1):12606.

38. Graham J, Özener B. Fluctuating Asymmetry of Human Populations: A Review. Symmetry. 2016;8(12).

39. Zhao L, Matloff W, Shi Y, Cabeen RP, Toga AW. Mapping Complex Brain Torque Components and Their Genetic and Phenomic Architecture in 24,112 healthy individuals. 2021.

40. Sha Z, Schijven D, Carrion-Castillo A, Joliot M, Mazoyer B, Fisher SE, et al. The genetic architecture of structural left-right asymmetry of the human brain. Nat Hum Behav. 2021.

41. Wachinger C, Golland P, Kremen W, Fischl B, Reuter M, Alzheimer's Disease Neuroimaging I. BrainPrint: a discriminative characterization of brain morphology. Neuroimage. 2015;109:232-48.

42. Wachinger C, Salat DH, Weiner M, Reuter M, Alzheimer's Disease Neuroimaging I. Whole-brain analysis reveals increased neuroanatomical asymmetries in dementia for hippocampus and amygdala. Brain. 2016;139(Pt 12):3253-66.

43. Richards R, Greimel E, Kliemann D, Koerte IK, Schulte-Korne G, Reuter M, et al. Increased hippocampal shape asymmetry and volumetric ventricular asymmetry in autism spectrum disorder. Neuroimage Clin. 2020;26:102207.

44. Ge T, Reuter M, Winkler AM, Holmes AJ, Lee PH, Tirrell LS, et al. Multidimensional heritability analysis of neuroanatomical shape. Nat Commun. 2016;7:13291.

45. Wachinger C, Nho K, Saykin AJ, Reuter M, Rieckmann A, Alzheimer's Disease Neuroimaging I. A Longitudinal Imaging Genetics Study of Neuroanatomical Asymmetry in Alzheimer's Disease. Biol Psychiatry. 2018;84(7):522-30.

46. Finn ES, Shen X, Scheinost D, Rosenberg MD, Huang J, Chun MM, et al. Functional connectome fingerprinting: identifying individuals using patterns of brain connectivity. Nat Neurosci. 2015;18(11):1664-71.

47. Mansour LS, Tian Y, Yeo BTT, Cropley V, Zalesky A. High-resolution connectomic fingerprints: Mapping neural identity and behavior. Neuroimage. 2021;229:117695.

48. Amico E, Goni J. The quest for identifiability in human functional connectomes. Sci Rep. 2018;8(1):8254.

49. Germanaud D, Lefevre J, Toro R, Fischer C, Dubois J, Hertz-Pannier L, et al. Larger is twistier: spectral analysis of gyrification (SPANGY) applied to adult brain size polymorphism. Neuroimage. 2012;63(3):1257-72.

50. Mars RB, Passingham RE, Jbabdi S. Connectivity Fingerprints: From Areal Descriptions to Abstract Spaces. Trends Cogn Sci. 2018;22(11):1026-37.

51. Noble S, Spann MN, Tokoglu F, Shen X, Constable RT, Scheinost D. Influences on the Test-Retest Reliability of Functional Connectivity MRI and its Relationship with Behavioral Utility. Cereb Cortex. 2017;27(11):5415-29.

52. Fischl B, Salat DH, Busa E, Albert M, Dieterich M, Haselgrove C, et al. Whole Brain Segmentation Automated Labeling of Neuroanatomical Structures in the Human Brain. Neuron. 2002;33(3):341–55.

53. Fischl B. FreeSurfer. Neuroimage. 2012;62(2):774-81.

54. Cardinale F, Chinnici G, Bramerio M, Mai R, Sartori I, Cossu M, et al. Validation of FreeSurfer-estimated brain cortical thickness: comparison with histologic measurements. Neuroinformatics. 2014;12(4):535-42.